# DAVE 🧑🏾‍🔬: Diagnostic benchmark for Audio Visual Evaluation

**Gorjan Radevski**[*]
KU Leuven
gorjan.radevski@kuleuven.be

**Teodora Popordanoska**[*]
KU Leuven
teodora.popordanoska@kuleuven.be

**Matthew B. Blaschko**
KU Leuven
matthew.blaschko@kuleuven.be

**Tinne Tuytelaars**
KU Leuven
tinne.tuytelaars@kuleuven.be

## Abstract

Audio-visual understanding is a rapidly evolving field that seeks to integrate and interpret information from both auditory and visual modalities. Despite recent advances in multi-modal learning, existing benchmarks often suffer from strong visual bias – when answers can be inferred from visual data alone – and provide only aggregate scores that conflate multiple sources of error. This makes it difficult to determine whether models struggle with visual understanding, audio interpretation, or audio-visual alignment. In this work, we introduce DAVE (Diagnostic Audio Visual Evaluation), a novel benchmark dataset designed to systematically evaluate audio-visual models across controlled settings. DAVE alleviates existing limitations by (i) ensuring both modalities are **necessary** to answer correctly and (ii) decoupling evaluation into atomic subcategories. Our detailed analysis of state-of-the-art models reveals specific failure modes and provides targeted insights for improvement. By offering this standardized diagnostic framework, we aim to facilitate more robust development of audio-visual models.

🤗 **Dataset:** https://huggingface.co/datasets/gorjanradevski/dave
⬛ **Code:** https://github.com/gorjanradevski/dave

## 1 Introduction

Large language models (LLMs) [Achiam et al., 2023, Chung et al., 2024, Touvron et al., 2023a,b, Bai et al., 2023] have demonstrated remarkable proficiency in understanding and generating text. Building on this success, recent work has extended these capabilities to other modalities through multi-modal LLMs (MLLMs) [Yin et al., 2023, Dai et al., 2023, Li et al., 2023b, Liu et al., 2024, Lyu et al., 2023, Maaz et al., 2024, Team et al., 2024, Su et al., 2023]. These models are designed to process and combine information from multiple sources, including images, audio, and video. Despite the rapid growth of MLLMs, there is still a significant lack of benchmarks tailored to rigorously evaluate their multimodal integration abilities.

Existing benchmarks for MLLM evaluation [Li et al., 2023a, Liu et al., 2025, Xu et al., 2024, Yu et al., 2023] predominantly focus on vision-language tasks, overlooking crucial multimodal interactions, such as audio-visual understanding and integration. Notably, current audio-visual datasets like MUSIC-AVQA [Li et al., 2022] and AVQA [Yang et al., 2022] are inadequate for assessing true

---

[*]Equal contribution.

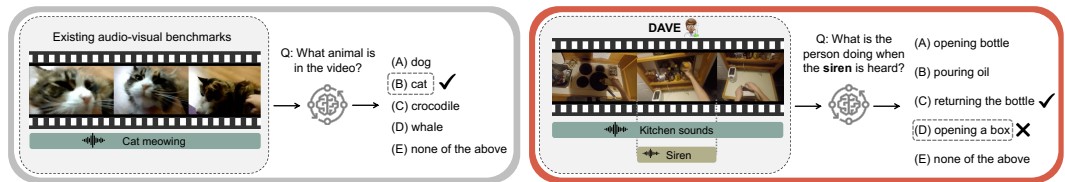

Figure 1: Existing benchmarks (e.g., AVQA [Yang et al., 2022]) suffer from visual bias (left) while DAVE (right) contains questions which are *impossible* to solve without both modalities.

multimodal alignment, as they exhibit a strong visual bias (see Fig. 1): i.e., their questions frequently permit answers to be derived from the visual modality itself. This fundamental limitation compromises the validity of performance metrics on these benchmarks, as high scores may reflect proficiency in unimodal comprehension rather than multimodal integration capabilities (see Fig. 2). A particularly significant gap exists in evaluating whether models understand the temporal alignment between visual and auditory events: for instance, determining if a sound occurs at the precise moment relative to an action event in a video. Without robust evaluation of this capability, it remains unclear whether models are genuinely integrating information across modalities or merely exploiting isolated cues from a dominant modality. Moreover, audio-visual comprehension encompasses multiple distinct cognitive subtasks that vary in complexity: sound event detection, action recognition, temporal alignment between audio and visual stimuli, cross-modal integration, and question comprehension. Single-metric evaluations obscure this complexity, potentially masking critical weaknesses in a model's ability to meaningfully synthesize information from multiple modalities. To accurately assess model performance, we argue that it is essential to decompose evaluation along these constituent dimensions, enabling more precise identification of failures.

In response to these challenges, we introduce **DAVE (Diagnostic benchmark for Audio Visual Evaluation)**, specifically designed to evaluate *audio-visual LLMs* (AV-LLMs). DAVE is designed around a key principle: each question requires information from both audio and visual modalities simultaneously, ensuring that neither modality *alone* is sufficient. To derive DAVE, we employ a novel semi-automatic data generation paradigm to generate multiple-choice questions and answers by leveraging Epic Kitchens [Damen et al., 2022] and Ego4D [Grauman et al., 2022] datasets. Using DAVE, we conduct comprehensive evaluations of several state-of-the-art AV-LLMs across several tasks. Our results reveal significant limitations in current models' ability to perform genuine multimodal comprehension, particularly with respect to temporal alignment and cross-modal integration.

To summarize, our main **contributions** are:

① We introduce DAVE, the first benchmark specifically designed to evaluate audio-visual synchronization and true multimodal understanding of AV-LLMs.

② We propose a decomposition of audio-visual reasoning into constituent subtasks, enabling fine-grained analysis of model performance beyond aggregate metrics.

③ We comprehensively evaluate several state-of-the-art AV-LLMs using our benchmark, uncovering limitations in multimodal integration and providing actionable insights for future model development.

## 2 Related work

The integration of multiple modalities, such as text, audio, and visual data, has become a key focus in the development of large language models. Recent progress in this field has been driven by improvements in model architectures, synthetic data generation, and training strategies.

In particular, **AV-LLMs** have made significant progress in bridging the gap between speech, environmental sounds, and visual information. A common approach in the development of MLLMs is aligning ImageBind [Girdhar et al., 2023] – a joint embedding space for multiple modalities – with an LLM to enable multimodal instruction-following. For instance, Panda-GPT [Su et al., 2023] integrates multimodal encoders from ImageBind [Girdhar et al., 2023] with Vicuna LLMs [Chiang et al., 2023], enabling instruction-following and emergent cross-modal integration across six modalities. Similarly, ImageBind-LLM [Han et al., 2023] aligns LLaMA with ImageBind's

joint embedding space via a learnable bind network, enabling multi-modality instruction-following across images, audio, 3D point clouds, and video. Beyond ImageBind-based approaches, several works utilize independently trained modality-specific encoders. Since the learned representations of different modalities may not be directly compatible, these approaches often focus on finding effective ways to align the multimodal representations. For example, X-InstructBLIP [Panagopoulou et al., 2023] aligns image, 3D, audio, and video to a frozen LLM, demonstrating emergent discriminative reasoning without modality-specific pretraining. Video-SALMONN [Sun et al., 2024] introduces a multi-resolution Q-Former to align time-synchronized audio-visual inputs with text. CAT [Ye et al., 2024] aggregates question-related audio-visual clues to improve the clarity of responses. Macaw-LLM [Lyu et al., 2023] jointly learns to align multi-modal features with LLM embeddings, by combining representation alignment and instruction tuning into a single step. VideoLLama2 [Cheng et al., 2024] integrates spatial-temporal modeling and audio processing, and achieves strong performance across video and audio-visual tasks. Meerkat [Chowdhury et al., 2024] introduces an audio-visual LLM with fine-grained spatio-temporal understanding. Some models unify multiple modalities in a single framework (e.g., One-LLM [Han et al., 2024]) using a universal projection module and progressive alignment across eight modalities, eliminating the need for modality-specific encoders. In contrast, we introduce a new audio-visual dataset and benchmark several of these models.

**Audio-visual QA datasets.** While numerous AV-LLMs have been proposed, the availability of benchmark datasets to systematically evaluate their capabilities remains limited. This scarcity hinders comprehensive analysis and comparison across models, slowing progress in the field. Effective evaluation requires benchmarks that assess a model's ability to understand, interpret, and integrate multimodal information. In the domain of audio-visual question answering (AVQA), several datasets have been proposed to support model training and evaluation.

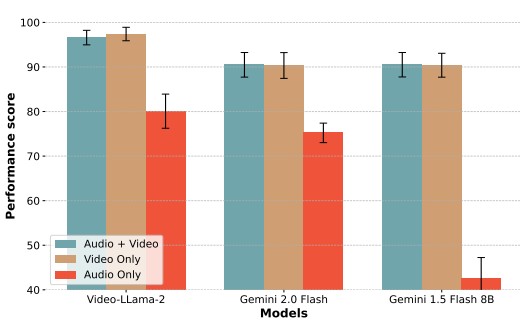

Figure 2: Performance comparison of AV-LLMs on AVQA [Yang et al., 2022] with different modalities. AVQA's questions exhibit a strong visual bias: performance using Video Only and Audio+Video input is consistently similar. Error bars represent standard deviations.

MUSIC-AVQA [Li et al., 2022] is a large-scale AVQA dataset, designed to evaluate spatio-temporal reasoning. It contains over 45K QA pairs generated from 33 templates, covering a collection of 9K musical performance videos. However, its focus on music-related scenes limits its generalizability, as the questions primarily explore instrument relationships. Thus, AVQA [Yang et al., 2022] was introduced with a more diverse set of audio-visual objects and activities drawn from everyday life. Despite this, most questions can be answered using only visual information (see Fig. 2), allowing models to succeed without truly integrating audio and visual modalities. Pano-AVQA [Yun et al., 2021] focuses on panoramic videos, however, its question types remain largely restricted to existence and spatial localization, with a limited focus on more complex relationships. *In contrast to these datasets, our benchmark is built around a key principle: each question is deliberately constructed to be impossible to solve using a single modality.* More recently, AVTrustBench [Chowdhury et al., 2025] introduced an audio-visual benchmark comprising 600K multiple-choice QAs over 9 tasks. It evaluates the trustworthiness of AV-LLMs in three areas: adversarial attack, compositional reasoning, and modality-specific dependency. In comparison, DAVE evaluates *audio-visual synchronization* capabilities in MLLMs, and is built on top of two large-scale (egocentric) datasets: Epic Kitchens [Damen et al., 2022] and Ego4D [Grauman et al., 2022]. By requiring genuine audio-visual integration, we establish a more demanding benchmark pushing models toward true multimodal understanding.

## 3  DAVE 🧑🏻‍🔬: Diagnostic benchmark for Audio Visual Evaluation

We focus on a multiple-choice AVQA problem, where $n$ answer options are provided, and adopt the "single-correct" setup, ensuring that only one of the options is correct. We present DAVE, a novel benchmark consisting of 2426 carefully curated samples, designed to evaluate audio-visual synchronization capabilities in MLLMs. A summary of the dataset statistics is shown in Fig. 3.

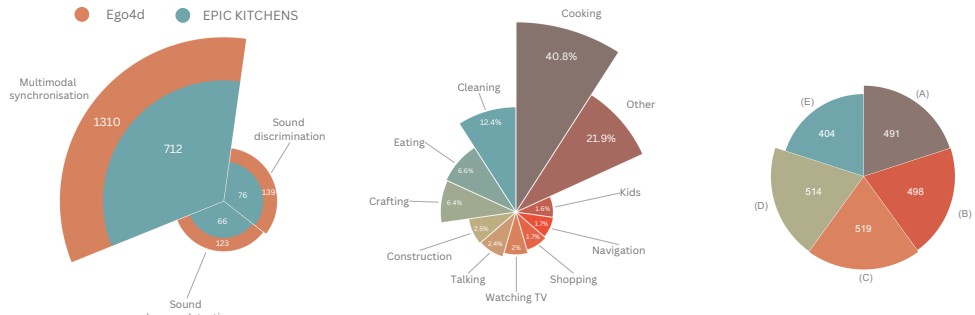

Figure 3: Dataset statistics. **Left:** Overview of the number of samples in DAVE 🧑🏽‍🔬, consisting of **2426** samples across three tasks and two source datasets. **Middle:** The 10 most common scenarios (around 78% of the data) in our benchmark. **Right:** Distribution of labels.

## 3.1 Data generation pipeline

**Data sources and preprocessing.** DAVE is built on top of two established (egocentric) video understanding datasets: Epic Kitchens [Damen et al., 2022] and Ego4D [Grauman et al., 2022]. These datasets provide natural human actions with rich temporal annotations, serving as an ideal foundation for audio-visual alignment tasks. For audio events, unrelated to the ones present in Epic Kitchens and Ego4D, we use the ESC-50 dataset [Piczak, 2015], comprising 2,000 environmental sounds across 50 classes. We select 13 audio classes and all their respective audio files. See App. A.2 for details.

**Event selection and grouping.** We define an *event* as a temporally localized segment containing a narrated action (e.g., "opening a drawer"). The event extraction process follows these steps:

(i) For each video, we extract the events, ensuring that we preserve the temporal boundaries, narrations, and action labels, which we then chronologically sort and merge consecutive events with identical actions to avoid redundancy.

(ii) We filter the events using several criteria: minimum and maximum duration threshold ($\tau_{min}$ and $\tau_{max}$) to ensure sufficient length of each event; [2] maximum overlap constraint ($\omega_{max}$) between consecutive events; narration quality filters to remove ambiguous descriptions where we remove events which contain specific words (see App. A.3 for details).

(iii) Qualified events are organized into *event groups*, each containing four sequential events that form a coherent activity sequence.

**Multimodal sample generation.** The procedure to generate audio-visual samples consists of:

(i) *Video processing.* Each event group is extracted as a continuous video segment while preserving the original audio track. Thus, the temporal coherence of the original video is maintained.

(ii) *Audio overlay.* Within each event group, we select events exceeding a minimum duration threshold ($\tau_{overlay}$) specified in seconds, for audio augmentation. We randomly select an environmental sound class (e.g., dog bark, door knock) for each qualified event. The selected audio is precisely aligned with the event's temporal boundaries to ensure synchronization between visual actions and auditory cues. We apply fade-in / fade-out effects ($\delta_{fade}$) to create natural transitions at the beginning and the end of the overlay. To maintain perceptual balance, we carefully adjust the volume ratio between original and overlaid audio using a scaling coefficient ($\alpha_{scale}$). See App. A.4 for details.

This procedure creates a dataset where temporal alignment between visual actions and synthetic audio events is precisely controlled, enabling rigorous evaluation of audio-visual integration capabilities.

**Data filtering and enhancing actions diversity.** To ensure high-quality samples and enhance diversity, we implement a two-stage filtering pipeline:

(i) *Narration enhancement and similarity filtering.* Event narrations in egocentric video datasets often contain abbreviated references (e.g., "C opens drawer") that lack natural language fluency. We employ an LLM (Gemini Flash 2.0 Lite) to rephrase these narrations into more human-readable descriptions. This process replaces camera-wearer references ("C") with natural alternatives ("The person" or "They") while preserving the original meaning. Additionally, we identify and filter out

---

[2]We observed that if the events are too short, it is difficult to recognize what action the person is performing.

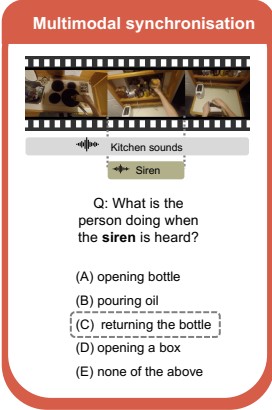
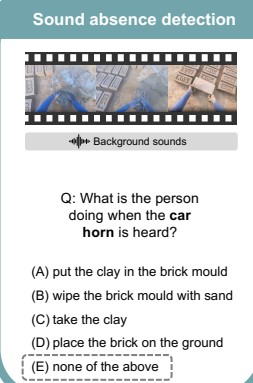
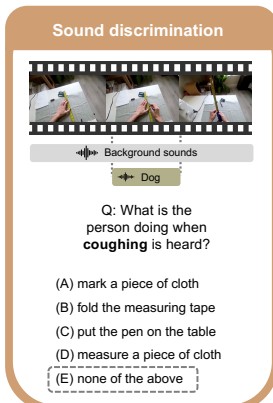

Figure 4: Illustration of the multimodal tasks in DAVE 🧑‍🔬. **Left:** Multimodal synchronisation tests if models can correctly identify actions occurring simultaneously with a specific sound (e.g., siren). **Center:** Sound absence detection evaluates if models can recognize when a queried sound (e.g., car horn) is absent. **Right:** Sound discrimination assesses if models can distinguish between different sound types (e.g., distinguishing between dog and coughing sounds) and avoid incorrect associations. Each task is multi-choice (correct answer with dashed line).

event groups with highly similar actions that could introduce ambiguity into our evaluation. Using the same language model, we compute a similarity score across narrations within each event group (see App. A.5 for details). Event groups exceeding a predetermined similarity threshold ($\tau_{sim}$) between the narrations are filtered out, ensuring that each remaining group contains only events with sufficiently distinct actions – crucial for creating effective distractors in our questions.

(ii) *Visual quality verification.* In the second stage, we address potential visual ambiguity issues. Even with clear narrations, some video segments may not depict the described action due to occlusion, poor lighting, or camera movement. To identify such cases, we employ Gemini Flash 2.0 Lite[3] to perform zero-shot action classification on each event segment. We frame this as a multiple-choice question answering task: for each event, we present the model with the video segment and four possible action categories from the four-event sequence, asking it to classify the action being performed (see prompt in App B.2). We then compare this classification against the ground truth narration. Events where the model fails to correctly identify the action are flagged as potentially ambiguous or unclear and filtered out from the final dataset.

This two-stage filtering process significantly improves the quality of our benchmark by: (i) ensuring linguistic clarity through enhanced narrations, (ii) maintaining semantic diversity within event groups, and (iii) verifying visual clarity of actions, thereby eliminating samples that could introduce noise into model evaluation. The resulting dataset provides a reliable foundation for evaluating audio-visual integration capabilities, as it minimizes ambiguity that could confound performance analysis. This quality-focused approach is particularly important for diagnostic benchmarks such as ours, where the goal is to isolate specific reasoning capabilities.

## 3.2 Question types

To effectively diagnose the multimodal understanding abilities of AV-LLMs, we decompose audio-visual reasoning into three distinct subtasks and design question types that specifically target each dimension (see Fig. 4). The multiple-choice format enables clear quantitative evaluation while requiring genuine cross-modal understanding.

**Multimodal synchronisation.** We test whether models can correctly link sounds to visual events happening at the same moment in time, see Fig. 4 (left). This requires the model to precisely align what it *sees* with what it *hears* – a fundamental skill for multimodal understanding. Given a video with an overlaid sound effect of a specific type, over a precisely segmented event associated with an action, we ask the models: What [action] takes place when [sound effect] is heard? For an AV-LLM

---

[3]Note that Gemini Flash 2.0 Lite is not used in the experiments we conduct.

to answer this question correctly, it must recognize the audio interval when the sound occurs and link that sound to the corresponding visual event – the person performing the action.

**Sound absence detection.** We evaluate if models can recognize when a sound is not present, preventing them from hallucinating non-existent audio-visual connections, as shown in Fig. 4 (middle). This tests the models' ability to avoid false positives in multimodal integration. Given a video *without any* overlaid unrelated sound (i.e., the video is presented in its original form), we prompt the model with the same question format as in multimodal synchronization. However, in this case, the model needs to determine that the mentioned sound is absent and correctly answer "none of the above".

**Sound discrimination.** We assess whether models can tell different sounds apart and avoid mixing them up, as in Fig. 4 (right). This helps us understand if models can correctly discriminate between similar audio cues rather than making incorrect associations. We provide the models with a video with an overlaid sound, but ask about a different sound that is not present in the video. This tests whether models detect *any* distinctive audio artifacts or truly understand audio semantics and correctly associate specific sounds with temporal video events.

### 3.3 Atomic tasks

In addition to the primary audio-visual understanding tasks (§3.2), we analyze three complementary atomic tasks designed to isolate and evaluate specific components of audio-visual understanding. See appendices B.1 to B.4 regarding the tasks' setup (i.e., how the audio-visual models are prompted).

**Action recognition task.** This task evaluates models' ability to recognize individual actions, independent of multimodal integration. Models are presented with isolated event segments cropped from the event group. Four multiple-choice options are provided, consisting of event narrations from the same event group. Models must classify the action occurring in the segment.

**Temporal ordering task.** This task assesses temporal reasoning abilities without requiring audio processing. Models view the complete event group video without the overlaid audio, and must correctly order a shuffled list of events based on their temporal sequence. This tests the model's ability to track and sequence actions in time, independent of audio cues.

**Audio discrimination task.** This task isolates audio perception capabilities. Models are presented with the full audio segment extracted from the video, including the overlaid sounds. Four multiple-choice options are provided, comprising a random subset of audio classes used as overlays. Models must identify which distinctive sound is present, testing semantic understanding of audio.

By decomposing the multimodal task into these atomic components, with DAVE we enable precise diagnosis of model strengths and weaknesses across modalities. This approach reveals whether performance gaps in the main audio-visual integration tasks stem from fundamental limitations in unimodal recognition, temporal understanding, or true cross-modal integration capabilities.

## 4 Results and discussion

### 4.1 Experimental setup

**Evaluation metrics.** We measure accuracy across all setups. For the open-source models, we also use an LLM (Gemini 2.0 Flash Lite) to judge the scores (i.e., LLM-as-a-judge [Zheng et al., 2023]) for the scenarios where the tested model does not follow instructions and outputs the full answer in natural text (see App. C.1). We report the standard deviation obtained by a bootstrap procedure (resampling with replacement) with 1000 iterations.

**Models.** We evaluate three categories of models on DAVE to assess the current state of audio-visual integration capabilities across different architectures, discussed below.

(i) *Closed-source end-to-end models.* We test several large-scale AV-LLMs designed to jointly process video and audio inputs – various versions of Gemini (1.5 Pro, 1.5 Flash, 1.5 Flash 8B, 2.0 Flash, and 2.5 Flash[4]). See appendices B.1 to B.4 for the prompts we used.

(ii) *Closed-source pipeline models.* We develop a modular pipeline approach that separates audio and visual reasoning. First, an audio-specialized model processes the audio track to identify times-

---

[4]Gemini 2.5 Flash was released after the submission deadline and was added in the camera-ready version.

Table 1: Performance of multimodal models across DAVE question types. We report accuracy (%) on the complete DAVE benchmark alongside performance on individual question types: multimodal synchronisation, sound absence detection, and sound discrimination. This highlights model-specific strengths and weaknesses in audio-visual reasoning. The error bars show standard deviation.

| | DAVE | Multimodal synchronisation | Sound absence detection | Sound discrimination |
|---|---|---|---|---|
| Human | $84.74_{\pm 2.26}$ | $85.75_{\pm 2.43}$ | $79.47_{\pm 6.43}$ | $81.34_{\pm 6.14}$ |
| Random | $22.41_{\pm 0.85}$ | $22.21_{\pm 0.88}$ | $22.54_{\pm 3.05}$ | $23.29_{\pm 2.86}$ |
| *Closed-source models* | | | | |
| Gemini 2.5 Flash | $58.73_{\pm 1.04}$ | $59.15_{\pm 1.14}$ | $70.69_{\pm 3.38}$ | $44.21_{\pm 3.44}$ |
| Gemini 2.0 Flash | $50.81_{\pm 1.01}$ | $59.86_{\pm 1.09}$ | $8.50_{\pm 2.10}$ | $2.75_{\pm 1.10}$ |
| Gemini 1.5 Flash | $49.53_{\pm 1.04}$ | $54.41_{\pm 1.14}$ | $31.71_{\pm 3.37}$ | $19.08_{\pm 2.76}$ |
| Gemini 1.5 Flash 8B | $28.64_{\pm 0.93}$ | $30.66_{\pm 1.01}$ | $22.38_{\pm 3.03}$ | $14.85_{\pm 2.52}$ |
| Gemini 1.5 Pro | $46.20_{\pm 1.02}$ | $54.98_{\pm 1.10}$ | $1.03_{\pm 0.77}$ | $2.78_{\pm 1.14}$ |
| *Closed-source pipelines* | | | | |
| GPT-4o | $27.38_{\pm 0.94}$ | $17.34_{\pm 0.85}$ | $60.12_{\pm 3.50}$ | $93.09_{\pm 1.74}$ |
| Gemini 1.5 Flash Pipeline | $35.34_{\pm 1.65}$ | $35.46_{\pm 1.84}$ | $18.10_{\pm 4.78}$ | $49.89_{\pm 5.76}$ |
| *Open-source models* | | | | |
| PandaGPT | $18.82_{\pm 0.80}$ | $16.52_{\pm 0.81}$ | $30.05_{\pm 3.29}$ | $30.25_{\pm 3.26}$ |
| video-SALMONN | $17.12_{\pm 0.77}$ | $20.09_{\pm 0.90}$ | $3.19_{\pm 1.27}$ | $2.34_{\pm 1.04}$ |
| Video-LLama-2 | $31.31_{\pm 0.95}$ | $36.32_{\pm 1.07}$ | $4.79_{\pm 1.56}$ | $7.37_{\pm 1.84}$ |

tamps where specified sound events occur, or outputs "None" if the sound is absent. The frames corresponding to these timestamps are then extracted and passed to a Video-LLM, which uses the visual information to determine the final multiple-choice answer. We implement this pipeline using GPT-4o and Gemini 1.5 Flash. See appendices B.5 and B.6 for the prompts we use for these models.

(iii) *Open-source models.* We evaluate three open-source AV-LLMs: (i) Video-LLaMA-2 [Cheng et al., 2024], which extends LLaMA with video understanding capabilities; (ii) PandaGPT [Su et al., 2023], which combines language modeling with audio-visual perception; and (iii) video-SALMONN [Sun et al., 2024], which specializes in speech and audio-language modeling for multimodal understanding. For each, we follow the implementation procedure as described in their respective repository, and download the released checkpoints. [5]

**Human evaluation.** To gain an intuition about the difficulty of the dataset for humans, we conduct a small-scale evaluation with five participants not included in the project. The participants are presented with the same audio-visual inputs as the models and asked to make predictions (see App. E).

### 4.2 Performance across question types

In Table 1, we analyze model performance across question types in DAVE. Notably, we observe a substantial gap between the best model (Gemini 2.5 Flash at 58.73% overall) and human performance on this task ($\sim$ 85%). This gap highlights the considerable challenges in developing models that effectively integrate and reason across audio and visual modalities with human-like proficiency.

*Insight 1*. All models perform significantly worse than humans on DAVE across all question types.

Further, models like Gemini 2.0 Flash and 1.5 Pro perform relatively well on multimodal synchronization (59.86% and 54.98% respectively), however their accuracy drops dramatically when tasked with detecting sound absence (8.50% and 1.03%) and sound discrimination (2.75% and 2.78%). We hypothesize that these models have been trained to actively seek and identify audio-visual correlations, but lack the ability to recognize when such correlations are absent. In other words, these models are biased toward making positive associations, even when the evidence does not support it.

Notably, Gemini 2.5 Flash avoids these catastrophic failures with more balanced performance across question types (59.15%, 70.69%, 44.21%), suggesting architectural or training improvements that enhance audio processing robustness. Additionally, we find that the GPT-4o pipeline model scores

---

[5]Two additional models which are suited for the specific task are Meerkat [Chowdhury et al., 2024] and CAT [Ye et al., 2024]. However, the authors have not released the checkpoints for these models.

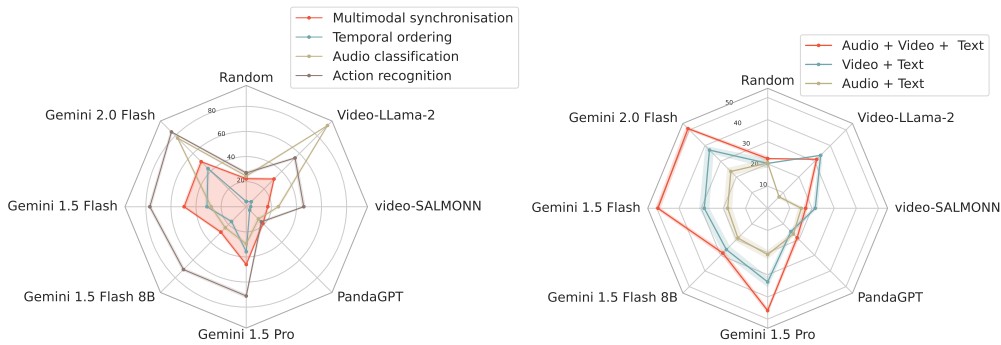

Figure 5: **Left.** Model performance on DAVE's composite task vs. atomic component tasks. We report accuracy (%) on the primary multimodal syncronisation task alongside performance on the constituent capabilities: temporal ordering, audio classification, and action recognition. This analysis reveals whether failures stem from weak component capabilities or true integration challenges (see Table 8). **Right.** Impact of modality availability on DAVE performance. We report accuracy when models have access to different modality combinations: full multimodal input (Audio + Video + Text), Video + Text, and Audio + Text. The performance degradation without all modalities demonstrates DAVE's effectiveness at requiring genuine cross-modal reasoning (see Table 9).

particularly well for the sound absence detection and sound discrimination tasks, as it frequently predicts "None of the above" (64.69% of samples) – a conservative strategy which is advantageous for this specific evaluation scenario.

The tendency to infer spurious audio-visual correlations represents a fundamental limitation in current multimodal architectures: the absence of explicit mechanisms to detect mismatches or missing correspondences between modalities. Addressing this limitation provides an actionable direction for improving model robustness: developing architectures with explicit "mismatch detection" modules or training objectives that reward correct identification of absent correlations.

> ***Insight 2***. Most models perform better at multimodal synchronization than sound absence detection and sound discrimination tasks.

## 4.3 Atomic task performance

We examine how models perform on the core functional capabilities needed for successful audio-visual integration, and report these results in Fig. 5 (left). By comparing the performance of the composite task against atomic subtasks, we can pinpoint whether failures stem from basic perception issues (i.e., action or audio recognition) or higher-level integration challenges.

We observe that models demonstrate substantially higher performance in recognizing actions visually than in performing multimodal integration. Across both closed-source and open-source models, we observe consistent performance gaps between action recognition and multimodal synchronization performance. For example, Gemini 1.5 Flash achieves 76.8% accuracy on action recognition (see Table 8), but only 49.51% on multimodal synchronization (a difference of around 25%). This disparity suggests that current models have developed robust visual action recognition capabilities in isolation but struggle significantly with the higher-order tasks of cross-modal integration. The bottleneck in multimodal understanding appears not to be in the perception of individual modalities but rather in integrating information across modalities with precise temporal correspondence.

This observation points to concrete improvement directions: future models should incorporate explicit temporal alignment modules or adopt training objectives that reward precise synchronization.

> ***Insight 3***. Models perform well at visual action recognition but struggle with multimodal integration.

Table 2: Multimodal synchronization accuracy (%) conditioned on atomic task success (✓). We report base synchronization performance and performance when conditioned on successful atomic tasks: audio classification (AC), action recognition (AR), temporal ordering (TO), and their combinations. This shows how improved atomic subtask performance affects overall multimodal capabilities. The values in subscript indicate the performance change relative to multimodal synchronization.

| Model | Multimodal synchronisation | Subtask-conditioned audio-visual performance | | | | |
|---|---|---|---|---|---|---|
| | | AC ✓ | AR ✓ | TO ✓ | AR & TO ✓ | AC & AR & TO ✓ |
| Gemini 2.5 Flash | 59.18 | $58.79_{-0.39}$ | $64.20_{+5.02}$ | $62.67_{+3.49}$ | $67.78_{+8.60}$ | $66.68_{+7.51}$ |
| Gemini 1.5 Flash | 54.40 | $55.12_{+0.72}$ | $61.82_{+7.41}$ | $54.42_{+0.01}$ | $59.41_{+5.00}$ | $64.96_{+10.56}$ |
| Gemini 1.5 Pro | 55.00 | $56.66_{+1.67}$ | $60.72_{+5.72}$ | $54.79_{-0.20}$ | $58.72_{+3.72}$ | $60.00_{+5.00}$ |
| Video-LLama-2 | 36.30 | $36.34_{+0.04}$ | $49.32_{+13.02}$ | $35.65_{-0.65}$ | $46.97_{+10.67}$ | $45.00_{+8.70}$ |

## 4.4 Modality ablation study

We investigate whether DAVE's questions genuinely require multimodal reasoning by conducting systematic ablation studies where we remove individual input modalities and measure the resulting performance changes across different model architectures. Unlike existing benchmarks that can be solved with single-modality shortcuts (see Fig. 2), we demonstrate that DAVE requires models to integrate information across audio, visual, and textual inputs. We visualize these results in Fig. 5 (right), while the full results are in Table 9.

The ablation study reveals that most models maintain above-random performance when restricted to video + text input, but experience significant performance degradation compared to the case when both audio and video are available.[6] For example, Gemini 2.0 Flash achieves 50.85% accuracy with full multimodal input but drops to 37.14% with only video and text. This demonstrates that DAVE effectively requires genuine multimodal integration.

> *Insight 4.* DAVE requires multimodal processing beyond single-modality cues for optimal performance.

## 4.5 Subtask-conditioned performance analysis

In Table 2, we explore the relationship between success on atomic tasks (see §3.3) and performance on audio-visual integration. This analysis helps us understand which atomic component capabilities contribute most to improved multimodal understanding.

We observe that models show substantially larger gains in audio-visual understanding from correctly identifying visual segments than from correctly classifying audio. Gemini 1.5 Flash improves by +7.41% with correct visual segments, but only +0.72% with correct audio classification, while Video-LLama-2 shows an even more prominent difference (+13.02% vs. +0.04%). This asymmetry suggests that video understanding serves as the primary foundation for effective multimodal integration.

Future model development should therefore prioritize improving visual action recognition and temporal alignment capabilities, as gains in audio classification alone yield minimal improvements without strong visual grounding.

> *Insight 5.* Visual understanding is more critical than audio classification for audio-visual performance.

## 5 Conclusion

We introduced DAVE, a benchmark specifically designed to evaluate audio-visual understanding capabilities in multimodal models. Unlike existing benchmarks that suffer from modality bias, DAVE explicitly requires information from both auditory and visual modalities, ensuring that neither modality alone is sufficient for correctly answering questions. Our comprehensive evaluation of

---

[6]We hypothesize that the better-than-random performance when using video and text only (without audio) stems from a dataset bias. Models seem to exploit a correlation between event length and audio presence, favoring longer video segments that more frequently contain the target audio. This systematic preference for longer clips over shorter distractors enables models to succeed without integrating audio and visual information.

state-of-the-art Audio-Visual LLMs reveals several critical insights. First, *all current models perform significantly below expected human performance*, highlighting the considerable challenges in developing systems that can effectively integrate and reason across modalities. Second, even though some models demonstrate reasonable capability in multimodal synchronization, *they struggle with sound absence detection and sound discrimination*, suggesting a bias toward making positive associations even when evidence does not support it. Furthermore, our decomposition of audio-visual reasoning into constituent subtasks reveals that *models excel at isolated visual action recognition but struggle with cross-modal temporal integration*. The substantial performance gap between multimodal and unimodal conditions confirms that DAVE *requires genuine multimodal integration*, validating it as a robust evaluation benchmark. Our findings emphasize the need for improved training strategies and architectures that can better handle temporal alignment between modalities, negative evidence reasoning, and sound discrimination. We hope that DAVE serves as a valuable diagnostic tool for the community, fostering progress in the development of more robust and truly multimodal learning systems that can approach human-level capabilities in cross-modal understanding.

**Limitations and future work.** While DAVE provides a structured and systematic evaluation of audio-visual models, it has certain limitations. First, our benchmark relies on a template-based question generation approach, which ensures consistency, but may limit the linguistic diversity and complexity of the questions compared to naturally occurring human-annotated questions. Second, DAVE is focused on multiple-choice question-answering tasks, which offers clear, structured evaluation metrics, but does not encompass all possible multimodal tasks, such as open-ended generation and causal reasoning. Expanding beyond multiple-choice formats could provide a more comprehensive understanding of model capabilities, and is left for future work. Third, we only evaluate a limited set of audio-visual models, whereas many more are present in the literature [Chowdhury et al., 2024, Chen et al., 2023b,a, Ye et al., 2024, Han et al., 2024, Su et al., 2023, Lyu et al., 2023, Panagopoulou et al., 2023, Cheng et al., 2024, Sun et al., 2024, Wang et al., 2024]. However, the models we evaluated represent a diverse selection of the ones that have publicly available checkpoints [Cheng et al., 2024, Su et al., 2023, Sun et al., 2024]. Finally, we build DAVE on top of datasets that feature only egocentric videos, while future works could extend such setup to videos of any type.

## Acknowledgments

This research received funding from the Research Foundation - Flanders (FWO) through project G0G2921N and the Flemish Government under the "Onderzoeksprogramma Artificiële Intelligentie (AI) Vlaanderen" programme. The resources and services used in this work were provided by the VSC (Flemish Supercomputer Center), funded by FWO and the Flemish Government. We are grateful to Dina Trajkovska, Viktor Gagaleski, Andrej Popordanoski, Simona Ivanova Ignjatovikj, Ivana Tushevska Tasevska, Dusan Grujicic and Dragan Simic for their help in establishing the human baseline for DAVE.

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

# Supplementary Material
# DAVE 🧑🏾: Diagnostic benchmark for Audio Visual Evaluation

The supplementary material is organized as follows:

- Details about the data generation procedure (App. A).

- The prompts for each task: audio-visual, action recognition, audio classification, temporal ordering, timestamp extraction, and event classification (App. B).

- Additional experiments and results on DAVE (App. C).

- Qualitative examples of DAVE question types (App. D).

- Details about the human performance analysis (App. E).

- Discussion of the societal impact (App. F).

## A  Data generation

### A.1  Datasets

Our dataset is constructed using the following publicly available datasets, each used in accordance with its respective license and terms of use:

- **Epic Kitchens** [Damen et al., 2022]: published under the Creative Commons Attribution-NonCommercial 4.0 International License (CC BY-NC 4.0).

- **Ego4D** [Grauman et al., 2022]: provided under a non-exclusive, non-transferable license for academic and research purposes (Ego4D-License).

- **ESC-50** [Piczak, 2015]: available under the terms of the Creative Commons Attribution Non-Commercial license (CC BY-NC 3.0).

### A.2  Audio classes selection

From the ESC-50 dataset [Piczak, 2015], we keep only the following sound events: "dog", "crow", "clapping", "chainsaw", "church bells", "clock alarm", "car horn", "laughing", "crying baby", "coughing", "sneezing", "siren", "cat". We chose these classes intentionally for the following reasons:

- Quality over quantity - We prioritized (and manually selected) audio classes that are clearly distinguishable and contextually appropriate for the scenarios in Epic Kitchens and Ego4D. This ensures high-quality diagnostic samples rather than a broad but potentially noisy audio.

- Specificity over breadth - Our primary goal is to evaluate audio-visual integration capabilities rather than comprehensive audio recognition. The selected classes provide sufficient diversity to test temporal alignment, sound discrimination, and absence detection without introducing unnecessary complexity.

### A.3  Words filtering

In order to ensure that the event descriptions are unambiguous, we filter all events, and thus, event groups, which contain one of the following words: "move", "moves", "moving", "unsure", "woman", "man", "person", "lady", "talks", "talk", "walk", "walks", "stand", "stands", "converses", "something", "look", "looks", "hold", "unknown", "fix", "fixes", "adjusts", "adjust", "stares", "turns", "turn", "hand".

These terms were excluded because they tend to be vague, overly generic, or refer to abstract or non-visual concepts, which might lead to ambiguity in the context of audio-visual (egocentric) video understanding.

## A.4 Implementation details

We intentionally use synthetically overlaid sounds that could naturally co-occur in real-world scenarios. For example, a person opening a fridge while coughing, or collecting water outdoors while a car horn sounds in the distance, are plausible everyday situations. We carefully selected audio classes from ESC-50 to ensure plausibility despite the synthetic overlay.

To construct high-quality audio-visual event pairs, we applied carefully selected duration thresholds and audio processing parameters. In particular, we empirically determine the minimum and maximum duration thresholds based on several considerations:

- We want to retain as many events as possible, hence, a too high minimum duration threshold, or too low maximum duration threshold will inevitably discard many events.

- If the minimum duration threshold is too low, we might end up with many events where it is difficult to determine the action the person is performing (Page 4, footnote 2). On the other hand, if the maximum duration threshold is too high, events may vary widely in length, which could introduce biases – e.g., models could default to predicting the longest action. Keeping event durations within a constrained range helps prevent the model from relying on length as a cue.

- We measure the average audio duration (without silent regions) in ESC-50, which we find to be 3.54 seconds.

Based on these considerations, we set event durations between $\tau_{\min} = 1.5$ and $\tau_{\max} = 10.0$ seconds, with full event groups limited to 60.0 seconds. We used a minimum overlay duration of $\tau_{\text{overlay}} = 3.5$ seconds, maximum event overlap of $\omega_{\max} = 0.5$ seconds, and an audio start offset of 1.5 seconds. These thresholds were selected empirically to maximize event retention, while ensuring sufficient action clarity and balanced segment lengths for fair multimodal evaluation.

For audio processing, we applied fade in/out effects of $\delta_{\text{fade}} = 0.3$ seconds and set the audio scale coefficient to $\alpha_{\text{scale}} = 1.3$, with each audio overlay processed to create natural transitions while ensuring synthetic sounds are clearly discernible within the natural audio context. We filtered narrations by removing events with ambiguous words (details below).

## A.5 Narration enhancement and similarity filtering

Since some of the narrations in the Ego4D dataset contain ambiguous forms such as "C open's the fridge", we rephrase the narrations using an LLM. Namely, we use a Gemini 2.0 Flash Lite model, and we prompt it in the following way:

> You are an advanced AI trained to process event narrations and make them more human-readable while assessing their similarity.
> **Task**
> - You will be given a list of short event descriptions.
> - In these descriptions, "C" refers to the camera wearer, who is also the person performing the action.
> - Your task is to rewrite each event description in a natural, human-readable way.
> - Avoid using "C" in the rephrased output. Instead, use "The person," "They," or rewrite the sentence naturally.
> - Additionally, analyze the provided narrations as a group and assign a similarity score to the set, reflecting how similar the events are to each other in terms of meaning.
> - If the events are highly distinct, assign a low similarity score. If they are highly similar or ambiguous, assign a high similarity score.
> **Output Format**
> Provide a JSON object with the following keys:
> - "score": A float value representing the overall similarity of the event descriptions.
> - rephrased_narrations: A list of strings where each event is rewritten to be more natural and readable.
> - Ensure that all output follows correct JSON syntax.

```
  - Use only standard double quotes (") for strings.
  - Do not include trailing commas or extra characters outside the JSON block. I need to load the
    output using json.loads
  **Guidelines for Rewriting**
  - Remove references to "C" and rewrite the event naturally.
  - Ensure clarity and readability.
  - Maintain the original event order and meaning.
  - Ensure that each event is distinct and easy to understand.
  - Use grammatically correct phrasing.
  **Event Descriptions**
  Here is a list of event descriptions:
  [list_of_events]
  Please rewrite them into a more human-readable format while maintaining their meaning.
  Additionally, analyze their similarity and assign a similarity score indicating how much the
  events resemble one another in meaning.
  Return the output in a RAW JSON format with the specified keys."""
```

We consider the risk of bias or hallucination in the rewriting process to be minimal due to:

- The rephrasing task is straightforward and rule-based: replacing "C" with "The person" or "They" and ensuring grammatical correctness. This could largely be achieved with simple string replacement, but we use an LLM for language fluency.

- The prompt contains eplicit instructions, output format constraints (JSON), and clear guidelines that minimize ambiguity (and thus potential for hallucination). Additionally, the prompt explicitly instructs to "maintain the original event order and meaning", preventing substantial alterations to the action descriptions.

- In a subsequent step, visually ambiguous events are filtered out using zero-shot action classification. To a large extent, we expect this to catch cases where rephrasing might have introduced inconsistencies with the visual action.

- The changes to the narration are primarily syntactic (pronoun replacement, grammatical corrections) rather than semantic, reducing the risk of introducing factual errors or biases.

Besides the rephrasing, the LLM returns a similarity score for the narrations themselves. In order to filter highly similar narrations (most likely belonging to the same human actions), we simply remove all instances for which the LLM score is above a threshold empirically set to 0.85. This filtering approach addresses specific quality issues that arise with action sequences that feature both highly similar and ambiguous narrations, which would create evaluation challenges. For example:

- "C picks a cloth" → "C puts the cloth in the washing machine" → "C picks a bedsheet" → "C puts the bedsheet in the washing machine"

- "C arranges them" → "C packs the spaghetti pieces" → "C packs another piece" → "C folds the spaghetti package"

- "C counts the papers" → "C puts some papers aside" → "C arranges the other papers" → "C measures the papers with the ruler"

Such sequences present several evaluation problems where the differences between actions like "picks a cloth" vs. "picks a bedsheet" or "packs the spaghetti pieces" vs. "packs another piece" are often too subtle or vague to create reliable multiple-choice distractors. Further, these contain repetitive patterns, such as (multiple "packs" or "arranges") that make it difficult to establish clear temporal boundaries. Finally, narrations like "arranges them" or "packs another piece" lack specificity, making it challenging to create meaningful questions that test genuine multimodal understanding rather than guessing.

# B  Prompts for each task

## B.1  Audio-visual task

What is the person doing when the [audio name] sound is heard in the background? Note that the sound might not be present in the video, in which case the correct answer would be 'None of the above'.
(A) [Action description 1]
(B) [Action description 2]
(C) [Action description 3]
(D) [Action description 4]
(E) none of the above
Answer only with the letter corresponding to your choice in parenthesis: (A), (B), (C), (D) or (E). Do not include any other text.

## B.2  Action recognition task

Prompt: Watch this short first-person (egocentric) video clip carefully. From the options below, select the action that most closely matches what the person is doing in the video. Choose the most appropriate option, even if it does not appear to be an exact match.
(A) [Action description 1]
(B) [Action description 2]
(C) [Action description 3]
(D) [Action description 4]
Answer only with the letter corresponding to your choice in parenthesis: (A), (B), (C) or (D). Do not include any other text.

## B.3  Audio classification task

Listen to the following audio clip carefully. In this clip, there are several environment sounds, but one of them is different or out of place. After listening to the audio, please identify which sound is not like the others. Choose the correct option from the list of multiple-choice answers below.
(A) [Audio class 1]
(B) [Audio class 2]
(C) [Audio class 3]
(D) [Audio class 4]
Answer only with the letter corresponding to your choice in parenthesis: (A), (B), (C) or (D). Do not include any other text.

## B.4  Temporal ordering task

The following are four actions that occur in a video. Your task is to order them based on their temporal sequence as they happen in the video.
(A) [Action description 1]
(B) [Action description 2]
(C) [Action description 3]
(D) [Action description 4]
Provide the sequence of letters that represents the correct temporal order. For example: (A)(B)(C)(D). Do not include any other text.

### B.5 Pipeline models: Timestamp extraction

> Please listen to the audio. There are several kitchen-environment sounds, but one ([audio name]) is different or out of place. Please identify the timestamp (start and end) when [audio name] occurs. Only output [timestamp start, timestamp end] in MM:SS format. Note that the sound might not be present at all, in which case only output 'None'.

### B.6 Pipeline models: Event classification

> These are frames from a video. What is the person doing in this video?
> (A) [Action description 1]
> (B) [Action description 2]
> (C) [Action description 3]
> (D) [Action description 4]
> Answer only with the letter corresponding to your choice in parenthesis: (A), (B), (C) or (D).
> Do not include any other text.

## C  Experiments

**Implementation details.**    To evaluate the Gemini [Team et al., 2024] and GPT-4o [Achiam et al., 2023] models, we utilize their respective official APIs. For Gemini models, we directly pass the full video sequence with overlaid audio sound, as they inherently support video inputs. In contrast, since GPT-4o does not currently support raw video inputs, we adopt a pipeline approach: we first prompt an audio model to identify the timestamps when a sound is heard, then extract the corresponding video frames and present them alongside a multiple-choice question to classify the depicted action. For the open-source models, we run inference on a single A100 GPU (40-80GB). We rely on the official GitHub repositories for the model checkpoints and implementation code:

- PandaGPT [Su et al., 2023] GitHub.
- video-SALMONN [Sun et al., 2024] GitHub.
- Video-LLama-2 [Cheng et al., 2024] GitHub.

In the following sections we report additional experiments.

### C.1  LLM-as-a-judge

Since the open source models we evaluate often do not follow the instructions, we evaluate them using the LLM-as-a-judge paradigm [Zheng et al., 2023]. For that, we use the Gemini Flash 2.0 Lite model and the following prompt:

> You will be given a multiple-choice question, the correct answer(s), and an LLM's response.
> Your task is to determine whether the LLM's response is correct.
> **Instructions:**
> - If the LLM's response matches any of the ground truth answers, return "Correct".
> - If the LLM's response does not match the ground truth, return "Incorrect".
> - Your response must be only "Correct" or "Incorrect" and nothing else.
> **Question:**
> prompt
> **Ground truth:**
> ground_truth
> **LLM output:**
> llm_output
> **Evaluation:**"""

Table 3: Performance of AV-LLMs across question types. We report LLM-as-a-judge accuracy (%) on the complete DAVE benchmark alongside performance on individual question types: multimodal synchronisation, sound absence detection and sound discrimination for the open source models.

| | DAVE | Multimodal synchronisation | Sound absence detection | Sound discrimination |
|---|---|---|---|---|
| **Open-source models** | | | | |
| PandaGPT | $19.12_{\pm 0.78}$ | $17.26_{\pm 0.82}$ | $31.17_{\pm 3.45}$ | $30.39_{\pm 3.01}$ |
| video-SALMONN | $17.78_{\pm 0.76}$ | $21.31_{\pm 0.86}$ | $3.18_{\pm 1.31}$ | $2.32_{\pm 1.03}$ |
| Video-LLama-2 | $32.26_{\pm 0.90}$ | $38.34_{\pm 1.05}$ | $5.69_{\pm 1.56}$ | $7.81_{\pm 1.78}$ |

In Table 3 we report the performance as measured with an LLM-as-a-judge method [Zheng et al., 2023], across different DAVE question types. The results indicate that performance remains largely consistent with previous evaluations, suggesting that using an LLM-as-a-judge does not significantly alter the assessment of model accuracy presented in the main paper.

## C.2 DAVE breakdown per source dataset

**DAVE question types.** In Tables 4 and 5 we report a detailed breakdown across DAVE question types, distinguishing performance across data samples sourced from Epic Kitchens and Ego4D, respectively. This split allows for a more granular analysis across different data distributions.

Table 4: Accuracy (%) of various AV-LLMs on the DAVE benchmark derived from the **Epic Kitchens** dataset across question types. This breakdown reveals model-specific strengths and weaknesses in different aspects of audio-visual reasoning.

| | DAVE | Multimodal synchronisation | Sound absence detection | Sound discrimination |
|---|---|---|---|---|
| Random | $20.88_{\pm 1.40}$ | $20.23_{\pm 1.50}$ | $21.40_{\pm 5.04}$ | $27.84_{\pm 5.02}$ |
| **Closed-source models** | | | | |
| Gemini 2.5 Flash | $62.06_{\pm 1.71}$ | $64.78_{\pm 1.86}$ | $67.49_{\pm 5.95}$ | $31.63_{\pm 5.39}$ |
| Gemini 2.0 Flash | $51.75_{\pm 1.76}$ | $62.25_{\pm 1.78}$ | $0.00$ | $1.29_{\pm 1.27}$ |
| Gemini 1.5 Flash | $51.69_{\pm 1.66}$ | $58.65_{\pm 1.84}$ | $24.23_{\pm 5.25}$ | $10.58_{\pm 3.58}$ |
| Gemini 1.5 Flash 8B | $30.66_{\pm 1.50}$ | $35.13_{\pm 1.77}$ | $11.97_{\pm 4.00}$ | $5.34_{\pm 2.62}$ |
| Gemini 1.5 Pro | $45.33_{\pm 1.62}$ | $54.16_{\pm 1.90}$ | $0.00$ | $2.67_{\pm 1.82}$ |
| Gemini 2.0 Flash Lite | $53.29_{\pm 1.67}$ | $52.21_{\pm 1.94}$ | $72.75_{\pm 5.45}$ | $44.61_{\pm 5.83}$ |
| **Closed-source pipelines** | | | | |
| GPT-4o | $33.21_{\pm 1.64}$ | $24.77_{\pm 1.63}$ | $58.80_{\pm 5.99}$ | $89.39_{\pm 3.53}$ |
| Gemini 1.5 Flash Pipeline | $35.41_{\pm 1.61}$ | $35.51_{\pm 1.77}$ | $18.17_{\pm 4.75}$ | $50.05_{\pm 5.86}$ |
| **Open-source models** | | | | |
| PandaGPT | $19.43_{\pm 1.39}$ | $17.12_{\pm 1.41}$ | $33.07_{\pm 5.83}$ | $29.12_{\pm 5.24}$ |
| video-SALMONN | $11.78_{\pm 1.09}$ | $12.79_{\pm 1.23}$ | $7.57_{\pm 3.25}$ | $5.21_{\pm 2.60}$ |
| Video-LLama-2 | $33.05_{\pm 1.63}$ | $39.58_{\pm 1.82}$ | $0.00$ | $2.62_{\pm 1.86}$ |

**DAVE composite tasks.** In Tables 6 and 7 we report a breakdown per dataset for the composite tasks: temporal ordering, audio classification, and action recognition. Overall, we observe that models tend to achieve higher performance on the Epic Kitchens subset compared to Ego4D. This trend likely stems from the higher quality of Epic Kitchens videos, which offer clearer visuals and better-curated narrations, making them easier for models to process. In contrast, Ego4D contains more diverse and less controlled footage, introducing additional challenges and leading to lower performance.

## C.3 DAVE subtasks and modalities

In Table 8 we show the performance of the various AV-LLMs on the defined subtasks, whereas in Table 9 we investigate the impact of the modality on DAVE performance. These tables serve as a more detailed overview of the results presented in Fig. 5.

Table 5: Accuracy (%) of various AV-LLMs on the DAVE benchmark derived from the **Ego4D** dataset across question types.

| | DAVE 🧑🏾 | Multimodal synchronisation | Sound absence detection | Sound discrimination |
|---|---|---|---|---|
| Random | $23.14_{\pm 1.06}$ | $23.36_{\pm 1.14}$ | $23.46_{\pm 3.76}$ | $20.90_{\pm 3.38}$ |
| *Closed-source models* | | | | |
| Gemini 2.5 Flash | $56.89_{\pm 1.27}$ | $56.17_{\pm 1.34}$ | $72.11_{\pm 4.18}$ | $51.35_{\pm 4.18}$ |
| Gemini 2.0 Flash | $50.29_{\pm 1.32}$ | $58.74_{\pm 1.39}$ | $12.93_{\pm 3.09}$ | $3.63_{\pm 1.58}$ |
| Gemini 1.5 Flash | $48.26_{\pm 1.23}$ | $52.06_{\pm 1.38}$ | $35.85_{\pm 4.18}$ | $23.52_{\pm 3.67}$ |
| Gemini 1.5 Pro | $46.71_{\pm 1.26}$ | $55.50_{\pm 1.33}$ | $1.60_{\pm 1.14}$ | $2.83_{\pm 1.41}$ |
| Gemini 1.5 Flash 8B | $27.44_{\pm 1.13}$ | $28.24_{\pm 1.22}$ | $27.57_{\pm 3.99}$ | $20.08_{\pm 3.38}$ |
| Gemini 2.0 Flash Lite | $43.27_{\pm 1.26}$ | $37.64_{\pm 1.31}$ | $81.34_{\pm 3.45}$ | $62.51_{\pm 4.15}$ |
| *Closed-source pipelines* | | | | |
| GPT-4o | $24.27_{\pm 1.07}$ | $13.31_{\pm 0.95}$ | $61.29_{\pm 4.28}$ | $95.01_{\pm 1.88}$ |
| Gemini 1.5 Flash Pipeline | $34.18_{\pm 1.18}$ | $34.62_{\pm 1.31}$ | $18.64_{\pm 3.53}$ | $43.23_{\pm 4.12}$ |
| *Open-source models* | | | | |
| PandaGPT | $18.39_{\pm 0.95}$ | $16.20_{\pm 1.02}$ | $28.45_{\pm 3.89}$ | $30.99_{\pm 4.07}$ |
| video-SALMONN | $20.14_{\pm 0.98}$ | $23.98_{\pm 1.21}$ | $0.80_{\pm 0.78}$ | $0.70_{\pm 0.69}$ |
| Video-LLama-2 | $30.31_{\pm 1.18}$ | $34.59_{\pm 1.35}$ | $7.33_{\pm 2.32}$ | $10.24_{\pm 2.54}$ |

Table 6: Model performance on DAVE's composite task versus atomic component tasks on the **Epic Kitchens**-based dataset.

| | Multimodal synchronisation | Temporal ordering | Audio classification | Action recognition |
|---|---|---|---|---|
| Random | $20.97_{\pm 1.42}$ | $2.69_{\pm 0.61}$ | $23.56_{\pm 1.58}$ | $31.30_{\pm 1.70}$ |
| *Closed-source models* | | | | |
| Gemini 2.5 Flash | $64.70_{\pm 1.82}$ | $60.63_{\pm 1.89}$ | $46.13_{\pm 1.85}$ | $85.29_{\pm 1.31}$ |
| Gemini 2.0 Flash | $51.99_{\pm 1.73}$ | $53.28_{\pm 1.80}$ | $76.57_{\pm 1.59}$ | $90.57_{\pm 1.13}$ |
| Gemini 1.5 Flash | $51.75_{\pm 1.71}$ | $32.88_{\pm 1.75}$ | $27.58_{\pm 1.67}$ | $83.56_{\pm 1.38}$ |
| Gemini 1.5 Flash 8B | $30.73_{\pm 1.58}$ | $20.07_{\pm 1.51}$ | $20.83_{\pm 1.53}$ | $77.95_{\pm 1.54}$ |
| Gemini 1.5 Pro | $45.32_{\pm 1.64}$ | $46.84_{\pm 1.92}$ | $29.01_{\pm 1.73}$ | $84.10_{\pm 1.38}$ |
| *Open-source models* | | | | |
| video-SALMONN | $11.71_{\pm 1.10}$ | $5.17_{\pm 0.85}$ | $26.34_{\pm 1.68}$ | $57.34_{\pm 1.87}$ |
| Video-LLama-2 | $33.14_{\pm 1.51}$ | $5.74_{\pm 0.87}$ | $91.10_{\pm 1.10}$ | $48.21_{\pm 1.87}$ |
| PandaGPT | $19.44_{\pm 1.33}$ | $5.20_{\pm 0.84}$ | $13.76_{\pm 1.29}$ | $11.08_{\pm 1.19}$ |

### C.4 Pipeline-based approaches

Finally, in Table 10 we investigate the performance of the pipeline-based approaches for solving audio-visual tasks. Specifically, we first prompt an audio model to extract the timestamps when a sound is heard in the overlaid audio. These timestamps are then used to extract corresponding video frames, which are subsequently processed by a video model to answer a multiple-choice question: "What is the person doing?". To assess the effectiveness of this approach, we evaluate performance on two key subtasks: action recognition and timestamp accuracy. Our results provide insights into how well each stage of the pipeline contributes to the final prediction. Note that we consider the timestamps (start and end) to be correctly predicted if the Intersection over Union (IoU) with the ground truth timestamps is greater than 0, i.e., at least some frames of the correct event will be provided to the video model for further processing.

## D Qualitative examples

In Fig. 6 we present qualitative examples from our proposed multiple-choice video question answering dataset DAVE. Each example illustrates four video frames extracted from a short video segment and paired with a specific sound-related question. The goal is to identify the correct action occurring when the sound is heard (multimodal synchronization), or to determine that no matching sound is

Table 7: Model performance on DAVE's composite task versus atomic component tasks on the **Ego4D**-based dataset.

| | Multimodal synchronisation | Temporal ordering | Audio classification | Action recognition |
|---|---|---|---|---|
| Random | $23.18_{\pm1.04}$ | $4.97_{\pm0.61}$ | $24.87_{\pm1.20}$ | $24.63_{\pm1.21}$ |
| *Closed-source models* | | | | |
| Gemini 2.5 Flash | $56.11_{\pm1.38}$ | $46.03_{\pm1.37}$ | $48.02_{\pm1.38}$ | $78.66_{\pm1.09}$ |
| Gemini 2.0 Flash | $50.30_{\pm1.24}$ | $37.64_{\pm1.31}$ | $77.90_{\pm1.15}$ | $80.93_{\pm1.11}$ |
| Gemini 1.5 Flash | $48.28_{\pm1.24}$ | $30.51_{\pm1.29}$ | $28.13_{\pm1.22}$ | $73.10_{\pm1.24}$ |
| Gemini 1.5 Flash 8B | $27.47_{\pm1.12}$ | $14.87_{\pm0.98}$ | $25.23_{\pm1.21}$ | $66.88_{\pm1.31}$ |
| Gemini 1.5 Pro | $46.59_{\pm1.22}$ | $35.82_{\pm1.32}$ | $29.58_{\pm1.26}$ | $71.01_{\pm1.28}$ |
| *Open-source models* | | | | |
| PandaGPT | $18.48_{\pm0.94}$ | $3.15_{\pm0.48}$ | $9.85_{\pm1.48}$ | $20.16_{\pm1.13}$ |
| video-SALMONN | $20.09_{\pm1.02}$ | $2.69_{\pm0.43}$ | $25.37_{\pm1.23}$ | $39.65_{\pm1.40}$ |
| Video-LLama-2 | $30.29_{\pm1.13}$ | $5.62_{\pm0.65}$ | $91.94_{\pm0.75}$ | $58.68_{\pm1.38}$ |

Table 8: Model performance on DAVE's composite task versus atomic component tasks. We report accuracy (%) on the primary multimodal syncronisation task alongside performance on the constituent capabilities: temporal ordering, audio classification, and action recognition. This analysis reveals whether multimodal integration failures stem from component perception deficiencies or true integration challenges. The table presents a more detailed overview of Fig 5 (Left).

| | Multimodal synchronisation | Temporal ordering | Audio classification | Action recognition |
|---|---|---|---|---|
| Random | $22.37_{\pm0.83}$ | $4.13_{\pm0.42}$ | $24.47_{\pm0.97}$ | $26.95_{\pm0.98}$ |
| *Closed-source models* | | | | |
| Gemini 2.0 Flash | $50.82_{\pm1.01}$ | $43.13_{\pm1.10}$ | $77.46_{\pm0.95}$ | $84.32_{\pm0.79}$ |
| Gemini 1.5 Flash | $49.51_{\pm1.00}$ | $31.31_{\pm1.05}$ | $28.02_{\pm1.02}$ | $76.84_{\pm0.94}$ |
| Gemini 1.5 Flash 8B | $28.61_{\pm0.88}$ | $16.64_{\pm0.83}$ | $23.64_{\pm0.95}$ | $70.77_{\pm1.00}$ |
| Gemini 1.5 Pro | $46.18_{\pm1.02}$ | $39.74_{\pm1.04}$ | $29.30_{\pm1.02}$ | $75.65_{\pm0.98}$ |
| *Open-source models* | | | | |
| PandaGPT | $18.81_{\pm0.78}$ | $1.86_{\pm0.42}$ | $13.79_{\pm1.27}$ | $17.02_{\pm0.85}$ |
| video-SALMONN | $17.17_{\pm0.77}$ | $3.56_{\pm0.41}$ | $25.63_{\pm0.98}$ | $45.85_{\pm1.13}$ |
| Video-LLama-2 | $31.27_{\pm0.91}$ | $5.71_{\pm0.53}$ | $91.60_{\pm0.61}$ | $54.99_{\pm1.11}$ |

present (sound absence detection or sound discrimination). For more qualitative examples, see the supplementary material where we provide 20 random samples from DAVE encompassing all three question types described in the main paper.

# E   Human performance

We present 5 people with the interface in Figure 7 and ask each to solve around 40 questions, without any further information about the task. Each person solves questions from the audio-visual task (analogous to Table 1). We note that this is not meant as a full-scale human study, as this is beyond the scope of this work, but rather a check to confirm that people can solve this task with high accuracy. The study involved only viewing publicly available video clips and answering multiple-choice questions. All participants were informed about the purpose of the study, the voluntary nature of their participation, and how their data would be used.

# F   Broader impact

In this work, we conduct a comprehensive analysis of state-of-the-art AVLLMs to examine their multimodal capabilities. Our study demonstrates that these models often struggle with audio-visual synchronization. By identifying these limitations, we aim to shed light on the potential

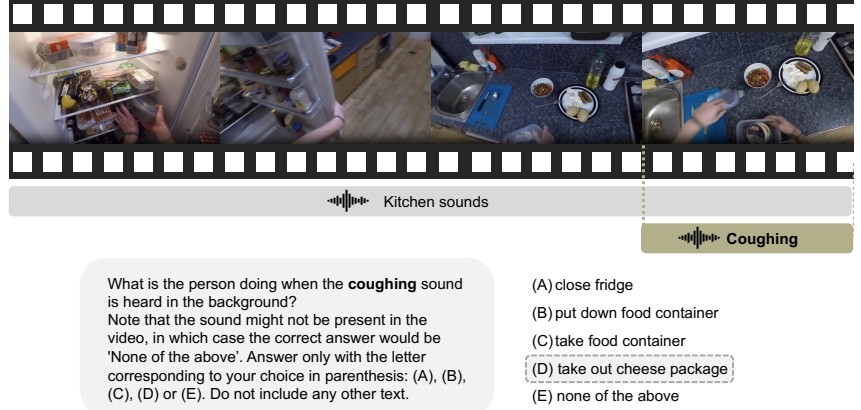

(a) Multimodal synchronisation

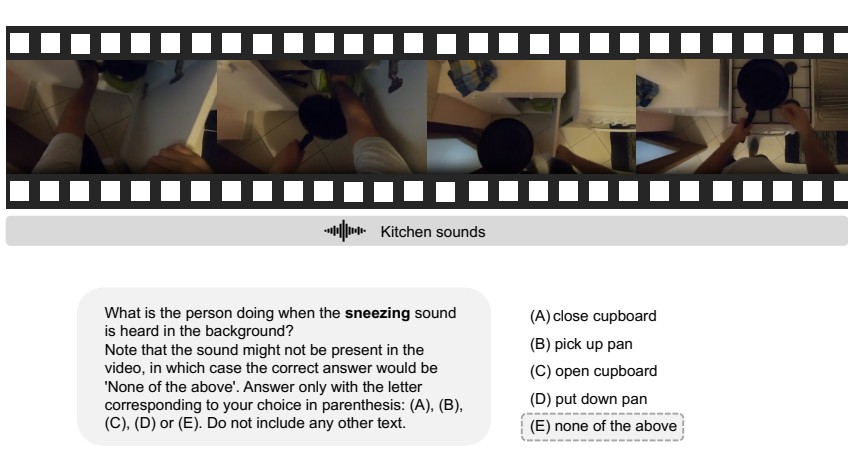

(b) Sound absence detection

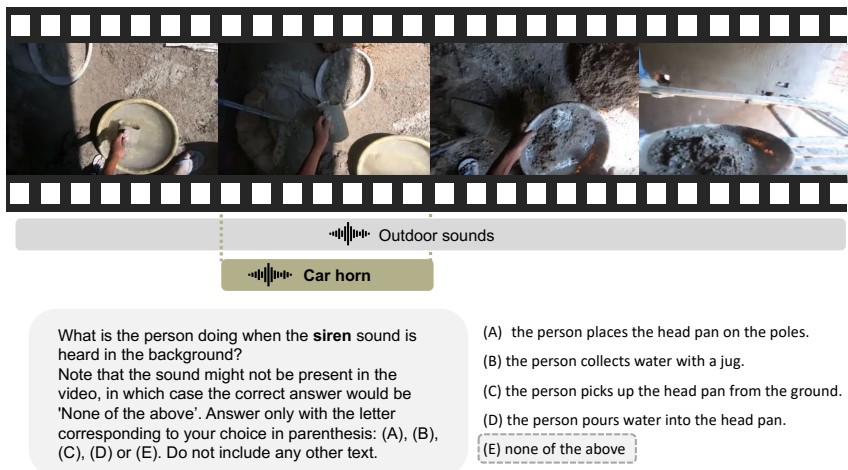

(c) Sound discrimination

Figure 6: Qualitative examples from our proposed multiple-choice video question answering dataset DAVE. Each example shows video frames corresponding to the four distinct events happening in the video segment. The task is to identify the action that occurs when the specified sound is heard (multimodal synchronization), or choose (E) "None of the above" in cases of sound absence (sound absence detection) or when a different sound is overlaid (sound discrimination).

Table 9: Impact of modality availability on DAVE performance. We report accuracy (%) when models have access to different modality combinations: full multimodal input (Audio + Video + Text), Video + Text only, Audio + Text only, and Text only. The performance degradation without all modalities demonstrates DAVE's effectiveness at requiring genuine cross-modal reasoning. The table presents a more detailed overview of Fig 5 (Right).

| | Audio + Video + Text | Video + Text | Audio + Text | Text |
|---|---|---|---|---|
| Random | $22.39_{\pm 0.85}$ | $20.25_{\pm 1.90}$ | $20.45_{\pm 1.79}$ | $20.09_{\pm 1.81}$ |
| *Closed-source models* | | | | |
| Gemini 2.0 Flash | $50.85_{\pm 1.00}$ | $37.14_{\pm 2.24}$ | $23.50_{\pm 1.97}$ | $17.83_{\pm 1.64}$ |
| Gemini 1.5 Flash | $49.51_{\pm 0.96}$ | $28.60_{\pm 2.05}$ | $18.04_{\pm 1.76}$ | $16.21_{\pm 1.68}$ |
| Gemini 1.5 Flash 8B | $28.61_{\pm 0.88}$ | $26.36_{\pm 1.95}$ | $18.98_{\pm 1.79}$ | $18.98_{\pm 1.78}$ |
| Gemini 1.5 Pro | $46.16_{\pm 0.99}$ | $33.25_{\pm 2.12}$ | $20.90_{\pm 1.80}$ | $16.51_{\pm 1.75}$ |
| *Open-source models* | | | | |
| PandaGPT | $18.80_{\pm 0.78}$ | $14.85_{\pm 0.74}$ | $16.42_{\pm 0.77}$ | – |
| video-SALMONN | $17.13_{\pm 0.75}$ | $21.47_{\pm 0.86}$ | $15.17_{\pm 1.31}$ | – |
| Video-LLama-2 | $31.27_{\pm 0.95}$ | $33.79_{\pm 1.05}$ | $7.28_{\pm 0.59}$ | – |

Table 10: Performance of the pipeline-based approach on DAVE. An audio model first extracts timestamps of the overlaid sound, which are used to select the corresponding video frames. Those frames serve as input to a video model, which answers the multiple-choice question: "What is the person doing in the video?". We report the accuracy of each stage. Note that we consider the timestamps (start and end) are correctly predicted if the IoU with the ground truth timestamps is $> 0$.

| | Multimodal | Action Recognition | Timestamp accuracy |
|---|---|---|---|
| GPT-4o | $27.39_{\pm 0.87}$ | $11.65_{\pm 0.71}$ | $40.10_{\pm 0.98}$ |
| Gemini 1.5 Flash Pipeline | $35.36_{\pm 1.64}$ | $43.05_{\pm 1.87}$ | $38.53_{\pm 1.67}$ |

risks and challenges associated with deploying these models in practical settings, where accurate synchronization between modalities is essential for reliable performance.

We believe our findings will be valuable to the community by highlighting areas where current models need improvement. Our proposed dataset for testing multimodal synchronization offers a resource for further evaluation and benchmarking of AV-LLMs, encouraging the development of models that are better equipped to handle the complexities of real-world data. In doing so, we hope to contribute to the advancement of more trustworthy and effective multimodal systems, promoting their safer and more responsible use in critical applications.

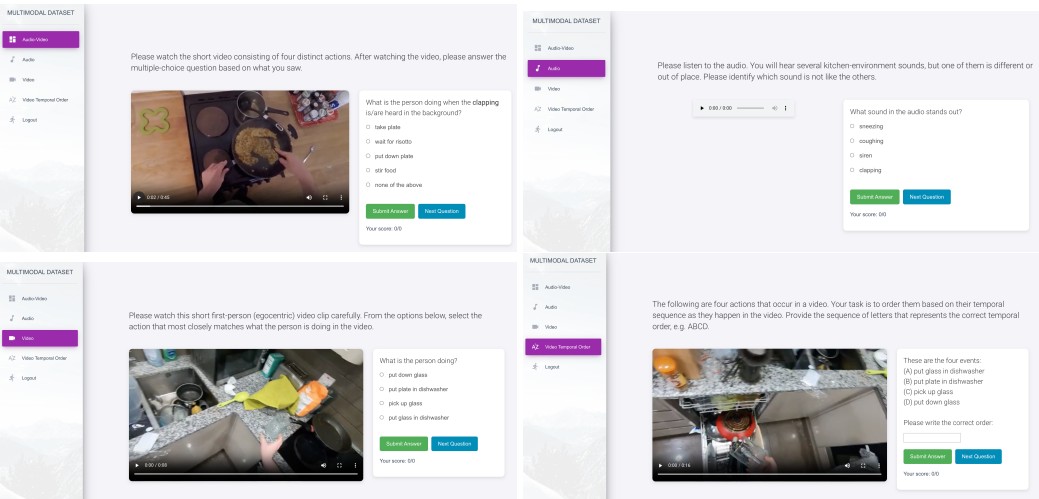

Figure 7: Web interface for evaluating human performance on the audio-visual task. Participants use this interface to answer multiple-choice questions without prior task-specific information, providing a baseline for comparison with model performance.

