# OpenReview forum: "DAVE: Diagnostic benchmark for Audio Visual Evaluation"
_NeurIPS.cc/2025/Datasets_and_Benchmarks_Track — NeurIPS 2025 Datasets and Benchmarks Track poster_

### Official Review · Reviewer_WBuA · 2025-06-06

**Rating:** 5
**Confidence:** 3

**Summary:**

This paper primarily focuses on audio-visual understanding and evaluates the effectiveness of multi-modal deep neural networks in terms of their cross-modal reasoning capabilities. The motivation for this research stems from the observation that existing benchmarks often suffer from strong visual biases (e.g., answers can be inferred from visual data alone) and typically report only aggregated scores, which conflate multiple sources of error. As a result, it becomes challenging to discern whether model performance issues arise from visual understanding, audio interpretation, or audio-visual alignment. To address these limitations, the authors propose DAVE (Diagnostic Audio-Visual Evaluation), a novel benchmark dataset designed to systematically assess audio-visual models under controlled conditions, with the goal of promoting the development of more robust audio-visual systems.

**Additional Feedback:**

- The Human performance on DAVE dataset it relatively lower than other benchmark (e.,g [VQA](https://visualqa.org/roe.html)). I am wondering what could be the reason behind this?

**Dataset Code Accessibility:**

Yes

**Ethical Considerations:**

No, there are no or only very minor ethics concerns

**Final Justification:**

In general this is a good paper for the field. Hence I keep my original score.

**Limitations Weaknesses:**

The 'Insight' box in the experimental section provides only a brief summary of the observed results, without offering deeper analysis or hypotheses related to the model development claims outlined in the third contribution. This may be considered a minor weakness of the paper.

**Strengths Contributions:**

- The paper is generally well-writtern and easy to follow the ideas and the motivation.
- A benchmark that can comprehensively evaluate the robustness of audio-visual models is crucial for the audio-visual research community, as most existing models suffer from visual bias.
- The proposed benchmark have three sub-categories for the model evaluation which can be benifited for the model evaluation.
- The paper provide sufficient visual examples in supplementary material for the qualitative visualisation.

---

> ### Author Rebuttal · Authors · 2025-07-30
>
> Thank you for your feedback and recognition of DAVE's importance to the audio-visual research community. We're pleased that you recognized the crucial need for a comprehensive benchmark to evaluate audio-visual model robustness, given the widespread visual bias in existing models. We're also encouraged that you appreciated our three evaluation sub-categories for more nuanced model assessment. We value your constructive suggestions and address your specific points below:
>
> > **The ‘Insight’ box briefly summarizes results but lacks deeper analysis related to the claims in the third contribution (Minor)**
>
> While the ‘Insight’ boxes in Section 4 are intentionally concise, each is grounded in quantitative results and observations presented in the surrounding text. As claimed in the third contribution (“we uncover limitations and provide actionable insights for model development”), several insights highlight limitations (e.g., performance gap between AV-LLMs and humans), while others offer concrete directions for improving cross-modal integration and robustness, specifically:
>
> - In Insight 2, we hypothesize that models are biased toward making positive audio-visual associations and lack mechanisms to recognize absence or mismatch between modalities. This offers a concrete direction for improving model sensitivity to negative evidence.
>
>
> - Insight 3 reveals a gap between unimodal perception (strong visual recognition) and temporal cross-modal integration, thus pinpointing a specific challenge (temporal integration) for future research to solve. This indicates that current architectures may benefit from inductive biases or training objectives focused on fine-grained temporal alignment.
>
>
> - Insight 5 states that "video understanding is more critical than audio classification", offering a priority for future model design (focus on robust visual understanding as a base for AV integration)
>
> We appreciate the reviewer’s suggestion and will revise the text to better connect the insights to concrete model development implications where appropriate.
>
>
> > **The Human performance on DAVE dataset is relatively lower than other benchmarks (e.g., VQA).**
>
> We appreciate the reviewer’s observation. While human performance on DAVE is slightly lower than certain question types in benchmarks like VQA (e.g., yes/no questions), it is important to consider the nature and complexity of the tasks. DAVE requires temporal and multimodal reasoning across four dynamic video segments, whereas VQA questions are based on a single static image and often involve shorter, more direct reasoning chains.
>
> Furthermore, note that the linked leaderboard shows better human performance only on the yes/no questions, whereas the overall average accuracy across all question types is around 80%, which is lower than the 85% we observe on DAVE.
>
> On our benchmark, based on the feedback from our human participants, the errors are due to:
>
> - Occasional lapses in attention, as each participant answered around 40 questions.
>
> - Rare edge cases in alignment, where the onset of the sound may occur during the transition between two actions, which can cause ambiguity.
>
> - Wording overlap: Infrequently, two answer choices are semantically equivalent  (e.g., “stir meat using spoon in saucepan” vs. “stir meat in saucepan using spoon”), but only one is marked as correct, leading to potential false negatives despite correct understanding.
>
> - A few participants misunderstood the “none of the above” cases, not realizing the absence or mismatch of sound was intentional, and chose incorrect answers despite the audio not matching.
>
> These issues are relatively infrequent, and overall, the 85% human accuracy demonstrates that the task is both feasible and well-posed, despite its higher complexity compared to traditional VQA.

---

### Official Review · Reviewer_WmDz · 2025-06-27

**Rating:** 5
**Confidence:** 3

**Summary:**

DAVE is a multiple-choice audio-visual QA dataset designed to pinpoint where AV-LLMs fail in integrating sound and vision. It comprises 2,426 carefully curated samples across three tasks—multimodal synchronization, sound absence detection, and sound discrimination—each ensuring that neither audio nor visual alone suffices for correct answers. A semi-automatic pipeline overlays precisely aligned environmental sounds, filters for linguistic clarity and visual quality, and rephrases narrations with an LLM to ensure human-readability and semantic diversity.

**Dataset Code Accessibility:**

Yes

**Dataset Code Comments:**

The authors provide dataset and code to make the benchmark accessible and reproducible.

**Ethical Considerations:**

No, there are no or only very minor ethics concerns

**Final Justification:**

1. Templates are a deliberate control. The authors isolate cross-modal skills, scale automatically, and give unambiguous answers, it is essential for rigorous, replicable metrics.

2. Coverage already extends beyond kitchens. Ego4D clips include crafting, construction, shopping, navigation, TV-watching, etc., so DAVE tests varied real-world contexts today.

I think this paper can be accepted.

**Limitations Weaknesses:**

1. Reliance on templated questions may limit linguistic diversity and fail to capture real-world query complexity; extending to open-ended prompts could offer richer evaluation.
2. Built exclusively on first-person views (Epic Kitchens, Ego4D), DAVE may not generalize to third-person or non-kitchen scenarios; broader video sources would increase ecological validity.

**Strengths Contributions:**

1. By decomposing audio-visual reasoning into atomic subtasks and three core QA types, DAVE reveals specific failure modes, unlike aggregate scores of prior benchmarks.
2. Ablation studies show performance drops when either modality is removed, demonstrating that DAVE’s questions require genuine cross-modal reasoning.
3. The two-stage filtering ensures unambiguous samples.
4. Benchmarks span closed-source end-to-end models (Gemini variants), pipeline approaches (GPT-4o + Video-LLM), and open-source AV-LLMs, with human baselines (~85% accuracy) highlighting a significant gap

---

> ### Author Rebuttal · Authors · 2025-07-30
>
> Thank you for your positive evaluation of our benchmark. We’re glad you recognized DAVE’s core strengths: genuine cross-modal reasoning, fine-grained decomposition to expose failure modes, ablation-based validation, rigorous filtering ensuring unambiguous samples, and comprehensive model benchmarking. Your acknowledgment of the human-model performance gap affirms the benchmark’s challenge and value.
>
> > **Reliance on templated questions may limit linguistic diversity**
>
> We agree with this point, as noted in our Limitations section. This design choice was intentional for several reasons:
>
> - Control: template-based questions make it easier to isolate specific multimodal reasoning skills. On the other hand, open-ended prompts introduce wide language variations and ambiguity, making it harder to pinpoint exactly what aspect of multimodal understanding the model is failing at.
>
> - Scalability: templates enable automatic, large-scale generation of questions without costly manual checking.
>
> - Evaluation: The fixed structure of templates ensures a consistent answer format (multiple-choice), which simplifies the quantitative evaluation.
>
> That said, we recognize the value of incorporating open-ended prompts in future work to capture richer language understanding.
>
> > **DAVE may not generalize to third-person or non-kitchen scenarios**
>
> As mentioned in our Limitations section, we agree that expanding to diverse video sources is a promising direction to enhance generalization. However, we’d like to note that Ego4D already contains a wide range of scenarios beyond kitchens – such as crafting, construction, watching tv, shopping, navigation, etc. This allows DAVE to already cover a more varied set of real-world contexts. Please refer to Figure 3 (middle) for the exact distribution of the top 10 scenarios in our benchmark.

---

> > ### Author Response · Authors · 2025-08-06
> > **Official comment**
> >
> > Dear Reviewer WmDz,
> > Thank you for your time and effort in reviewing our paper. We believe we have addressed all your questions and concerns. Since the deadline for the discussion period is approaching, please let us know if you have any remaining concerns or feedback.

---

### Official Review · Reviewer_bV91 · 2025-07-02

**Rating:** 4
**Confidence:** 4

**Summary:**

DAVE is an innovative Audio-Visual Evaluation benchmark, whose setup is designed to address the visual bias in existing datasets. To comprehensively evaluate the audio-visual alignment capability of models, DAVE ensures that both modalities are necessary for correct answers, while splitting the evaluation into multiple atomic tasks.

**Dataset Code Accessibility:**

Yes

**Ethical Considerations:**

No, there are no or only very minor ethics concerns

**Final Justification:**

The author's reply solved my problem.

**Limitations Weaknesses:**

- When using Gemini Flash 2.0 Lite for narrative rewriting, are there additional checking mechanisms? Is it possible for biases or hallucinations to be introduced due to the model's own reasons?
- Actions that are highly similar but not entirely identical seem to be valuable, as combining them with corresponding audio may further enable the checking of the performance of multimodal models.
- During the visual quality verification phase, Gemini Flash 2.0 Lite is used to perform action classification tasks. Are all categories provided in the prompt at this point?
- In the paper, incidents where Gemini Flash 2.0 Lite failed to correctly recognize actions were labeled as potentially ambiguous or unclear and thus excluded from the final dataset. This approach directly limits the upper bound of the dataset. In other words, if Gemini Flash 2.0 Lite exhibits poor classification performance on certain types of incidents, those incidents will be removed on a large scale.

**Strengths Contributions:**

- DAVE takes into account the issue of visual bias in existing datasets, so the core principle in its construction is: each question explicitly requires the simultaneous use of information from both audio and visual modalities, ensuring that information from a single modality is insufficient for answering.
- DAVE decomposes audio-visual reasoning into constituent atomic tasks, thereby enabling fine-grained analysis of model performance.
- DAVE comprehensively evaluates a variety of state-of-the-art audio-visual large language models, revealing the limitations of multimodal fusion.

---

> ### Author Rebuttal · Authors · 2025-07-30
>
> Thank you for your comprehensive and constructive feedback. We're pleased that you recognized DAVE's key contributions: (1) addressing visual bias by requiring simultaneous audio-visual information for each question, (2) decomposing audio-visual reasoning into atomic tasks for fine-grained analysis, and (3) comprehensively evaluating state-of-the-art models to reveal multimodal fusion limitations. We address your questions below.
>
> >**Are there additional checking mechanisms for Gemini Flash 2.0 Lite in narrative rewriting to prevent biases or hallucinations?**
>
> We consider the risk of bias or hallucination in the rewriting process to be minimal due to these factors:
> - The rephrasing task is straightforward and rule-based: replacing "C" with "The person" or "They" and ensuring grammatical correctness. This could largely be achieved with simple string replacement, but we use an LLM for language fluency.
> - We try to make the prompt (Appendix A.5) with explicit instructions, output format constraints (JSON), and clear guidelines that minimize ambiguity (and thus potential for hallucination). Additionally, the prompt explicitly instructs to "maintain the original event order and meaning," preventing substantial alterations to the action descriptions.
> - In a subsequent step, visually ambiguous events are filtered out using zero-shot action classification. To a large extent, we expect this to catch cases where rephrasing might have introduced inconsistencies with the visual action.
> -  The changes to the narration are primarily syntactic (pronoun replacement, grammatical corrections) rather than semantic, reducing the risk of introducing factual errors or biases.
>
> Finally, any potential biases in this preprocessing step would affect all models and the human equally; thus, the core diagnostic capabilities of our benchmark (testing temporal alignment and multimodal integration) remain unchanged.
>
> >**Highly similar but not identical actions seem valuable for multimodal model performance checking.**
>
> We appreciate this perspective on the value of similar actions. However, our filtering approach addresses specific quality issues that arise with action sequences that feature both highly similar and ambiguous narrations, which would create evaluation challenges.
>
> For example:
> - "C picks a cloth" → "C puts the cloth in the washing machine" → "C picks a bedsheet" → "C puts the bedsheet in the washing machine"
> - "C arranges them" → "C packs the spaghetti pieces" → "C packs another piece" → "C folds the spaghetti package"
> - "C counts the papers" → "C puts some papers aside" → "C arranges the other papers" → "C measures the papers with the ruler"
>
> Such sequences present several evaluation problems where the differences between actions like "picks a cloth" vs. "picks a bedsheet" or "packs the spaghetti pieces" vs. "packs another piece" are often too subtle or vague to create reliable multiple-choice distractors. Further, these contain repetitive patterns, such as (multiple "packs" or "arranges") that make it difficult to establish clear temporal boundaries. Finally, narrations like "arranges them" or "packs another piece" lack specificity, making it challenging to create meaningful questions that test genuine multimodal understanding rather than guessing.
> While we agree that subtle action differences could provide valuable evaluation scenarios, the similar actions we filter are cases where the narrations themselves are ambiguous or repetitive, rather than meaningfully distinct actions that could benefit from multimodal discrimination.
>
> > **Are all categories provided in the prompt during visual quality verification with Gemini Flash 2.0 Lite?**
>
> Yes, we frame the task as multiple-choice question answering and provide Gemini Flash 2.0 Lite with both the video excerpt and the four possible action categories from the 4-event sequence. The exact prompt is given in App. B.2. We will clarify this in the text.
>
> >**Excluding incidents where Gemini Flash 2.0 Lite failed action recognition directly limits the dataset's upper bound.**
>
> We acknowledge that filtering based on Gemini Flash 2.0 Lite's action recognition capabilities may introduce some bias. However, this approach serves important quality assurance purposes:
>
> - Events that cannot be visually identified even by a capable model likely suffer from occlusion or ambiguous visual cues that would compromise evaluation validity (e.g., the camera wearer may look upward or away while an action—such as placing an object on a counter or interacting with a drawer—occurs outside the field of view).
> - We prefer excluding potentially problematic samples rather than including unclear cases that could introduce noise and confound model comparisons. Our goal is diagnostic evaluation, which requires high-quality, unambiguous samples where performance differences reflect genuine multimodal capabilities rather than visual ambiguity.
>
> We are aware that this filtering may exclude some valid samples; however, the resulting benchmark provides better insight into the multimodal integration abilities of the models. We want to restate that the substantial performance gaps we observe (human: 84.74% vs. best model: 50.81%) suggest that current limitations stem from fundamental multimodal integration challenges rather than dataset coverage constraints.

---

> > ### Author Response · Authors · 2025-08-06
> > **Official comment**
> >
> > Dear Reviewer bV91,
> > Thank you for your time and effort in reviewing our paper. We believe we have addressed all your questions and concerns. Since the deadline for the discussion period is approaching, please let us know if you have any remaining concerns or feedback.

---

> > > ### Comment · Reviewer_bV91 · 2025-08-07
> > >
> > > Thanks to the author's response, I've corrected my score accordingly.

---

### Official Review · Reviewer_gBzk · 2025-07-03

**Rating:** 4
**Confidence:** 4

**Summary:**

DAVE is a new benchmark designed to evaluate audio-visual large language models (AV-LLMs) by requiring both audio and visual input for accurate answers. Unlike prior benchmarks with visual bias and aggregate metrics, DAVE decomposes evaluation into subtasks (e.g., sound detection, temporal alignment) to reveal specific model weaknesses. It uses data from Epic Kitchens and Ego4D to test true multimodal understanding.

**Dataset Code Accessibility:**

Yes

**Dataset Code Comments:**

Authors provided link to access dataset and code as well.

**Ethical Considerations:**

No, there are no or only very minor ethics concerns

**Final Justification:**

In author's rebuttal, they addressed most of my conerns, and therefore, I am increasing my rating.

**Limitations Weaknesses:**

Weaknesses:

- In line 140, the authors refer to a "minimum and maximum duration threshold," but they do not provide any explanation or details on how these thresholds were determined.
- In line 231, it is unclear how the authors assessed whether the scores correspond to scenarios where the tested model failed to follow instructions. Because as questions are multiple choose LLM will always return one choice.
- The number of participants involved in the human evaluation is quite limited. I would have expected a larger and more reasonable sample size for the experiment.
- In Tables 1 and 2, the evaluation metrics used to measure the performance across different tasks are not clearly specified. The names of the metrics are missing.
- It uses synthetically overlaid sounds rather than naturally co-occurring audio events. While this allows precise control, it may not fully reflect real-world audio-visual complexity, limiting ecological validity.
- The benchmark uses only 13 audio classes selected from the ESC-50 dataset, which may not represent the full range of real-world environmental sounds, possibly restricting generalizability.

**Strengths Contributions:**

Strengths:

- The paper is well-written and easy to follow.
- The authors provide qualitative results in the supplementary material, which is a valuable way to visualize and understand the dataset.
- DAVE explicitly requires both audio and visual information for correct answers, eliminating the visual bias present in previous benchmarks which is interesting.

---

> ### Author Rebuttal · Authors · 2025-07-30
>
> Thank you for the valuable feedback. We’re glad that you found DAVE’s core principle — requiring both audio and visual information for correct answers — to be a clear advancement over previous benchmarks suffering from visual bias. We’re also encouraged that you found the paper well-written and the supplementary qualitative results helpful for visualizing and understanding the dataset. We address your questions/concerns below:
> *
> >**How were the minimum and maximum duration thresholds determined? (Line 140)**
>
> We empirically determine the minimum and maximum duration thresholds based on several criteria:
>
> - We want to retain as many events as possible, hence, a too high minimum duration threshold, or too low maximum duration threshold will inevitably discard many events.
>
> - If the minimum duration threshold is too low, we might end up with many events where it is difficult to determine the action the person is performing (Page 4, footnote 2). On the other hand, if the maximum duration threshold is too high, events may vary widely in length, which could introduce biases – e.g. models could default to predicting the longest action. Keeping event durations within a constrained range helps prevent the model from relying on length as a cue.
>
> - We measure the average audio duration (without silent regions) in ESC-50, which we find to be ~3.54 seconds.
>
> Therefore, we set the minimum event duration to 1.5 seconds, the maximum event duration to 10.0 seconds, and the minimum length of the event that will be overlaid with synthetic audio to 3.5 seconds to match the ESC dataset. These thresholds were selected empirically to maximize event retention, while ensuring sufficient action clarity and balanced segment lengths for fair multimodal evaluation. We reported these values in Appendix A.4, and we will include a more detailed explanation for our choice.
>
> >**How was it assessed whether scores correspond to instruction-following failures when LLMs always return a choice? (Line 231)**
>
> We have observed that some of the open-source models we evaluate fail to follow instructions when prompted to return only a single letter corresponding to the choice. For example, in certain scenarios, instead of the model returning just the letter in parenthesis, e.g., “(A)”, the model produces variants such as: “\<A\>, “(A”, “A”,  “The answer is (A)”, “The answer is (A) close cupboard”, “ A. close cupboard”, or even “The answer is: close cupboard”. To address this variability and ensure fair evaluation, we additionally reported LLM-as-judge scores, where we prompt an LLM to determine if the prediction matches the ground truth (Appendix C.1).
>
> The results indicate that the LLM-as-a-judge scores (Table 3) remain relatively close to the scores obtained with our regex-based evaluation in Table 1. This similarity suggests that the evaluation results in the main paper are stable across different extraction methods.
>
> >**Limited sample size: "The number of participants in human evaluation is quite limited."**
>
> We appreciate the reviewer’s concern regarding the number of participants in our human evaluation; however, we want to clarify the purpose and scope of this evaluation within our study design.
>
> We design the human evaluation as a feasibility check rather than a comprehensive human study. We explicitly state this in our Appendix E: "We note that this is not meant as a full-scale human study, as this is beyond the scope of this work, but rather a check to confirm that people can solve this task with high accuracy."
>
> The primary goal is to establish that the task is solvable by humans with reasonable accuracy, in order to validate that DAVE is both a meaningful and a challenging benchmark. The human performance (84.74% accuracy) successfully demonstrates this.
>
> Importantly, we report error bars via bootstrap sampling with 1000 iterations in Table 1, which allows us to determine that there is indeed a statistically significant difference between human performance and the models’ performance across the different question types.
>
> A comprehensive human study with larger sample sizes would indeed be valuable for establishing robust human baselines. However, such an extensive study is beyond the scope of our current work, which focuses on benchmark development and model evaluation.
>
> >**Evaluation metrics in Tables 1 and 2 are not clearly specified.**
>
> We mention that we measure accuracy in the Experimental setup (Line 230), and we also state the same in the caption of Table 1. We’ll state the same in the caption of Table 2. We want to thank the reviewer for pointing this out to us.
>
> >**Use of synthetically overlaid sounds rather than naturally co-occurring audio events, limiting ecological validity.**
>
> We intentionally use synthetically overlaid sounds that could naturally co-occur in real-world scenarios. For example, a person opening a fridge while coughing, or collecting water outdoors while a car horn sounds in the distance, are plausible everyday situations. We carefully selected audio classes from ESC-50 to ensure plausibility despite the synthetic overlay.
> Importantly, a synthetic overlay provides essential advantages for rigorous evaluation:
>
> - Precise temporal control - we can ensure exact alignment between overlayed audio and visual events, which is necessary for testing temporal synchronization capabilities.
>
> - Controlled experimental conditions - we can systematically vary audio presence/absence/intensity and types to create diagnostic test cases (sound absence detection, sound discrimination).
>
> - Elimination of confounds - naturally occurring audio-visual pairs often contain multiple simultaneous sounds or ambiguous temporal boundaries that could confound evaluation.
>
> - Clear mapping – in our benchmark, each synthetic audio event corresponds precisely to a distinct visual action, avoiding the complexity where natural audio may span multiple actions, which complicates framing the task as multiple-choice question answering.
>
> Overall, we acknowledge that a synthetic overlay may not capture all real-world audio-visual complexity; however, this controlled approach is essential for diagnostic benchmarking. Our goal is to isolate and test specific multimodal integration capabilities rather than evaluate performance on naturalistic but potentially confounded scenarios.
>
> >**Only 13 audio classes from ESC-50 may not represent the full range of real-world environmental sounds.**
>
> We acknowledge that using 13 audio classes from ESC-50 represents merely a subset of the possible environmental sounds. However, we made this selection intentionally for a few reasons:
>
>  - Quality over quantity - We prioritized (and manually selected) audio classes that are clearly distinguishable and contextually appropriate for the scenarios in Epic Kitchens and Ego4D. This ensures high-quality diagnostic samples rather than a broad but potentially noisy audio.
>
>  - Specificity over breadth - Our primary goal is to evaluate audio-visual integration capabilities rather than comprehensive audio recognition. The selected classes provide sufficient diversity to test temporal alignment, sound discrimination, and absence detection without introducing unnecessary complexity.
>
> We agree that expanding to more audio classes would increase coverage, however, we want to state that the current set successfully demonstrates the significant limitations in existing models' multimodal integration abilities. The substantial performance gaps we observe between models and humans suggest that the challenge lies in fundamental multimodal reasoning rather than audio vocabulary breadth.

---

> > ### Comment · Reviewer_gBzk · 2025-08-05
> > **response to the rebuttal**
> >
> > The authors addressed all my concerns, and therefore, I am increasing my initial points to borderline accept.

---

> > > ### Author Response · Authors · 2025-08-06
> > > **Thank you**
> > >
> > > Thank you for revisiting our paper and for your updated evaluation. We're glad that our responses addressed all your concerns.

---

### Author Response · Authors · 2025-08-04
**General response**

**We thank all reviewers for their thoughtful and constructive feedback.**
DAVE was designed as an audio-visual benchmark where answering each question truly requires both audio and visual input, effectively avoiding shortcuts through unimodal cues (present in existing benchmarks such as AVQA).
We are encouraged that the reviewers consistently recognized our main contributions:

1. **Clear motivation and benchmark design**
- “DAVE explicitly requires both audio and visual information for correct answers, eliminating the visual bias present in previous benchmarks” (**gBzk**)
- “Each question explicitly requires the simultaneous use of information from both modalities” (**bV91**)
- The benchmark is “crucial for the audio-visual research community, as most existing models suffer from visual bias” (**WBuA**)
- “DAVE’s questions require genuine cross-modal reasoning” (**WmDz**)

2. **Fine-grained evaluation**

- “DAVE reveals specific failure modes, unlike aggregate scores of prior benchmarks” (**WmDZ**)
- “DAVE decomposes audio-visual reasoning into constituent atomic tasks, enabling fine-grained analysis of model performance” (**bV91**)

3. **Comprehensive benchmarking**

- “DAVE comprehensively evaluates a variety of state-of-the-art audio-visual large language models, revealing the limitations of multimodal fusion.” (**bV91**)
- “Benchmarks span closed-source end-to-end models (Gemini variants), pipeline approaches (GPT-4o + Video-LLM), and open-source AV-LLMs” (**WmDz**)

4. **Clarity and presentation**

- “Well-written and easy to follow” (**gBzk, WBuA**)
- “sufficient visual examples in supplementary material for the qualitative visualisation” (**WBuA**)
- “authors provide qualitative results in the supplementary material, which is a valuable way to visualize and understand the dataset” (**gBzk**)

Reviewers also raised several thoughtful questions, which we address in our individual responses. We summarize the main concerns below.

- **Synthetic audio overlay (gBzk)**: While gBzk raised concerns that synthetic sounds "may not fully reflect real-world audio-visual complexity," our design choice enables precise diagnostic evaluation. Synthetic overlay provides essential advantages: controlled temporal alignment for testing synchronization, systematic variation for diagnostic test cases, and elimination of natural audio confounds. We use plausible everyday scenarios (e.g., opening the fridge while coughing) to maintain real-world applicability.

- **Limited audio classes (gBzk)**: gBzk noted that using 13 ESC-50 classes "may not represent the full range of real-world environmental sounds." However, this represents a deliberate quality-over-quantity approach prioritizing clearly distinguishable, contextually appropriate sounds. The substantial human-model performance gap (84.74% vs 50.81%) demonstrates that current challenges stem from fundamental multimodal reasoning limitations rather than audio vocabulary breadth.

- **Data Curation Concerns (bV91)**:

    - **Narration rewriting:** bV91 questioned whether "biases or hallucinations" could be introduced when using Gemini for rewriting the answer choices (e.g. replacing “C” with “the person” or “they). The risk is minimal due to straightforward, rule-based rephrasing with explicit constraints that maintain original meaning.
    - **Action recognition**: While bV91 noted that filtering samples based on Gemini’s inability to recognize the action being performed "directly limits the upper bound of the dataset," we prioritize diagnostic quality over coverage. Events unrecognizable even by capable models likely suffer from ambiguity that would compromise evaluation validity.

- **Template-based questions (WmDz)**: WmDz identified that "reliance on templated questions may limit linguistic diversity and fail to capture real-world query complexity." We acknowledge this limitation in our paper, but note that templates enable controlled isolation of specific multimodal reasoning skills and scalable generation essential for diagnostic benchmarking.

- **Generalization beyond ego-centric scenarios (WmDz)**: WmDz noted DAVE is "built exclusively on first-person views" which "may not generalize to third-person or non-kitchen scenarios." We clarify that Ego4D encompasses diverse contexts beyond kitchens, including crafting, construction, entertainment, shopping, and navigation (Figure 3), providing substantial coverage of real-world scenarios.

**We look forward to additional feedback or constructive discussion for clarification.**

---

### Note · Authors · 2025-08-13

We thank all reviewers for their constructive feedback and for engaging in the discussion phase. We are encouraged by the consistent recognition of DAVE's core contributions: (1) eliminating visual bias through genuine cross-modal (audio-visual) understanding requirements, (2) enabling fine-grained analysis of multimodal integration failures, and (3) comprehensive benchmarking of state-of-the-art models.

The rebuttal period allowed us to clarify core design choices, including our controlled synthetic-audio overlay for precise diagnostic evaluation and our quality-over-quantity filtering strategy to ensure unambiguous, reliable samples.

We believe DAVE fills a critical gap in the audio-visual research community by offering the first benchmark that truly requires both modalities to answer correctly, and by exposing a substantial human-model performance gap that highlights fundamental limitations in current multimodal integration capabilities.

---

### Decision · Program_Chairs · 2025-09-18

**Decision:**

Accept (poster)

**Comment:**

## Summary:
DAVE introduces a diagnostic benchmark to evaluate audio-visual models by requiring both modalities and decomposing tasks into atomic subcategories (e.g., synchronization, sound absence detection). It identifies specific model weaknesses, such as poor cross-modal alignment and overreliance on visual cues.

## Strengths:

+ Addresses visual bias by design, ensuring both modalities are necessary.
+ Controlled synthetic audio-visual pairing and fine-grained subtasks isolate failure modes.
+ Public dataset and code with human baselines validating task feasibility.
+ Comprehensive evaluation of state-of-the-art models reveals substantial human-model performance gap (84.74% vs 50.81%).

## Weaknesses:

+ Synthetic audio limits ecological validity (mitigated by controlled evaluation needs).
+ Templated questions restrict linguistic diversity (trade-off for reproducibility).
+ Narrow audio classes (13 from ESC-50) but sufficient for diagnostic insights.

## Rebuttal Points:
+ Thresholds & Metrics (gBzk): Authors clarified empirical threshold selection and metric definitions (accuracy) and Justified small sample as a feasibility check with statistical significance.
+ Gemini Filtering (bV91): Authors explained safeguards against bias and prioritization of unambiguous samples.
+ Dataset Scope (WmDz): Authors highlighted Ego4D’s diverse scenarios (crafting, navigation) to counter generalizability concerns.
+ Insight Depth (WBuA): Authors expanded connections between results and model development implications (e.g., temporal integration challenges).

## Justification for Accept:
DAVE fills a critical gap in audio-visual research with rigorous, actionable diagnostics. The rebuttal resolved concerns (thresholds, Gemini filtering, human evaluation scope), and reviewers agreed on its technical soundness and community value.